# UniSS: Unified Expressive Speech-to-Speech Translation with Your Voice

**Sitong Cheng**[1], **Weizhen Bian**[1], **Xinsheng Wang**[2], **Ruibin Yuan**[1], **Jianyi Chen**[1],
**Shunshun Yin**[2], **Yike Guo**[1]*, **Wei Xue**[1]*
[1]Hong Kong University of Science and Technology
[2]Soul AI Lab
🌐Website: https://cmots.github.io/uniss-demo

## Abstract

The ultimate goal of expressive speech-to-speech translation (S2ST) is to accurately translate spoken content while preserving the speaker identity and emotional style. However, progress in this field is largely hindered by three key challenges: the scarcity of paired speech data that retains expressive styles, the complexity of multi-stage processing pipelines, and the limited transfer of translation capabilities from large language models (LLMs). In this work, we address these challenges by introducing UniSS, a novel single-stage framework for expressive S2ST. Our approach features carefully designed speech semantic and style modeling, enabling seamless integration with existing text-based LLM frameworks to develop a unified text-speech language model. To transfer translation capabilities from text to speech, we propose a cross-modal chain-of-thought prompting process that progressively aligns audio semantics with text and ensures style preservation in the decoded results. Furthermore, we construct and release a large-scale, high-quality expressive S2ST dataset, UniST, comprising 44.8k hours of data. Experimental results show that UniSS significantly outperforms previous methods in translation fidelity and speech quality while preserving voice, emotion, and duration consistency. Our work establishes a simpler and more effective paradigm for building the next generation of expressive S2ST systems. Audio samples are available at `https://cmots.github.io/uniss-demo/`.

## 1 Introduction

Speech-to-speech translation (S2ST) enables conversion from a spoken utterance in one language to a spoken utterance in another, facilitating critical applications like real-time interpretation and cross-lingual video dubbing. Recent advancements have significantly improved translation fidelity from a semantic perspective (Jia et al., 2019; Lee et al., 2022; Inaguma et al., 2023). However, achieving high-quality translation that also faithfully preserves the expressive aspects of speech, such as speaker identity and emotional style, still remains an open challenge.

A conventional cascaded S2ST system typically consists of three sequential components: automatic speech recognition (ASR), text-to-text machine translation (MT), and text-to-speech (TTS) synthesis (Casacuberta et al., 2004; Nakamura et al., 2006). However, this cascaded architecture often suffers from error accumulation across stages and struggles to retain paralinguistic features of the original speech. To address these limitations, subsequent research has shifted towards end-to-end approaches that aim to translate speech to another language while preserving expressive characteristics (Barrault et al., 2023; Wang et al., 2024; Le et al., 2024).

Most recently, the application of large language models (LLMs) to generative speech tasks has further accelerated the development of S2ST systems. Current approaches typically fall into two categories: (1) single-stage methods, which directly predict multi-stream acoustic tokens autoregressively via multi-head outputs (Labiausse et al., 2025); and (2) two-stage pipelines, which first generate semantic tokens autoregressively, followed by another autoregressive (AR) model to predict acoustic tokens (Dong et al., 2024; Rubenstein et al., 2023). Gong & Veluri (2024) and Peng et al. (2024) use a single AR language model to jointly model semantic and partial acoustic tokens, along with a non-autoregressive (NAR) model for complete acoustic information. While these

---

*Corresponding authors.

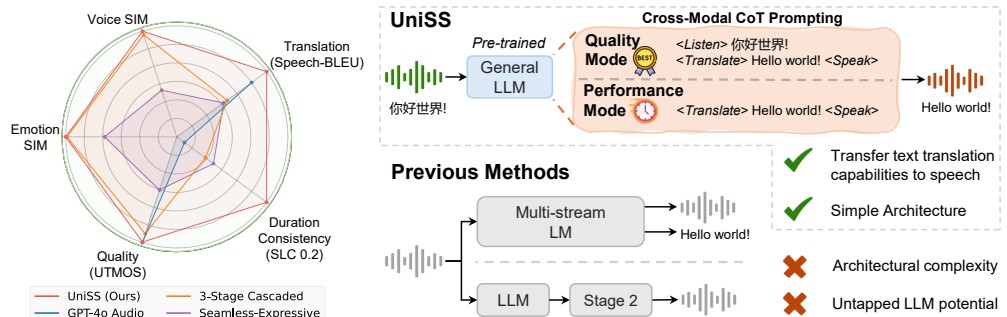

Figure 1: UniSS integrates pre-trained text LLMs and transfers their translation capabilities to the speech domain. UniSS outperforms both end-to-end S2ST systems and cascaded systems on translation fidelity, voice preservation, duration consistency, and speech quality.

approaches have shown promising results, they also introduce significantly more architectural complexity than textual LLMs. Additionally, they treat the LLM as a sequence-to-sequence converter, failing to leverage the pre-trained knowledge for textual translation embedded within the LLM.

In pursuit of high-fidelity, expressive S2ST, and to overcome the limitations outlined above, our vision is defined by three core principles: (1) a single-stage architecture to eliminate complexity; (2) a unified model that aligns speech and text modalities; and (3) a mechanism to explicitly leverage the proven text translation capabilities of LLMs. To our knowledge, no existing approach satisfies all three principles simultaneously.

To address the challenge of architectural complexity, we present **UniSS**, a unified single-stage S2ST framework that preserves timbre, emotion, and duration consistency. Figure 1 demonstrates our approach and performance results. UniSS builds upon pre-trained Qwen2.5-1.5B-Instruct (Yang et al., 2024) as the backbone, modeling content and speaker information in a single AR language model without architectural modifications. We transfer the LLM's pre-trained text translation capabilities to the speech domain through a cross-modality chain-of-thought (CoT) prompting (Wei et al., 2022). During inference, our Quality Mode guides the model to decompose the complex S2ST task into sequential listen, translate, and speak steps within a single inference pass. We also introduce a Performance Mode that trades translation fidelity for faster inference, enabling deployment across diverse scenarios.

Realizing the vision outlined above demands large-scale, high-quality training data that preserves both translation accuracy and speaker expressiveness. However, existing S2ST datasets are either small-scale to train powerful unified models (Jia et al., 2022b) or suffer from quality control issues when scraped from the web (Barrault et al., 2023). To address this data challenge, we design a scalable synthesis pipeline and contribute UniST, a 44.8k-hour Chinese-English S2ST dataset offering high translation fidelity and rich speaker preservation.

Our main contributions are as follows:

- **A Unified Single-Stage Architecture for S2ST:** We propose UniSS, a single-stage AR model for expressive S2ST that directly leverages a pre-trained textual LLM, eliminating the architectural complexity of prior work.
- **Transfer of Text Translation Capabilities to Speech:** We demonstrate how to effectively leverage pre-trained LLMs' superior text translation abilities for speech translation, bridging the modality gap through cross-modal CoT prompting and enabling flexible control over quality-efficiency trade-offs.
- **A Large-Scale, High-Quality S2ST Dataset:** We design a reproducible pipeline to create expressive S2ST datasets and construct UniST, a large-scale Chinese-English S2ST dataset with speaker voice and emotion preservation.

## 2 RELATED WORK

### 2.1 SPEECH-TO-SPEECH TRANSLATION

Speech-to-speech translation must preserve semantic accuracy during cross-lingual conversion. Traditional cascaded systems (Casacuberta et al., 2004; Nakamura et al., 2006) chain ASR, MT, and

TTS components sequentially, suffering from error accumulation and information loss through text bottlenecks. Early direct methods (Jia et al., 2019) encountered significant challenges with translation quality and synthesis artifacts. The breakthrough came with discrete unit-based methods, where speech-to-unit translation (S2UT) approaches (Lee et al., 2022; Zhang et al., 2021; Inaguma et al., 2023) employ discrete speech units (subwords, phonemes, or semantic tokens) as intermediate representations, which enables effective disentanglement of linguistic content from acoustic properties. SeamlessM4T (Barrault et al., 2023) achieved robust multilingual performance via unified multitask optimization of translation and synthesis. Recent systems focus on enhancing expressiveness while maintaining translation fidelity (Jia et al., 2022a; Song et al., 2023; Wang et al., 2024). SeamlessExpressive (Barrault et al., 2023) designs a PRETSSEL vocoder (Hwang et al., 2024) to enhance expressiveness preservation. TransVIP (Le et al., 2024) employs feature disentanglement to separately model semantic, acoustic, and temporal information for voice and isochrony control.

## 2.2 S2ST WITH AUTOREGRESSIVE LANGUAGE MODELS

The success of LLMs in text processing has inspired their adaptation to S2ST through speech tokenization. Wang et al. (2023) and Hu et al. (2025) focus primarily on semantic translation without preserving voice characteristics. To address voice preservation, some works model acoustic features using multiple AR models (Rubenstein et al., 2023; Dong et al., 2024). Recent unified approaches attempt to model both semantic and acoustic features within a single AR model (Peng et al., 2024; Gong & Veluri, 2024). However, they still require an additional NAR model for complete acoustic generation. Hibiki (Labiausse et al., 2025) proposes a multi-stream architecture processing source and target speech simultaneously, but necessitates substantial architectural modifications, including nested Transformers and training from scratch. A fundamental limitation across these approaches is their reliance on complex architectures to complete acoustic representations. Moreover, they fail to explicitly leverage the LLM's pre-trained text translation capabilities, treating them merely as generic sequence converters. UniSS addresses both limitations through a single-stage architecture while effectively transferring LLM's text translation capabilities to speech.

## 3 METHOD

As illustrated in Figure 2, UniSS is a unified AR language model for expressive speech-to-speech translation, built upon a single-stage architecture with a cross-modal chain-of-thought prompting framework, and optimized via a progressive training strategy. This section details the three core components of UniSS: model architecture, cross-modal CoT prompting, and training methodology.

### 3.1 UNISS ARCHITECTURE

The goal of UniSS is to perform expressive S2ST by modeling the conditional distribution $P(\mathbf{Y}_{tgt}|\mathbf{X}_{src})$, where $\mathbf{X}_{src}$ and $\mathbf{Y}_{tgt}$ denote the source and target speech waveforms, respectively. To achieve faithful content translation while preserving speaker identity and expressiveness, UniSS introduces three types of speech tokens: speaker tokens $\mathbf{S}^{spk}$ with fixed sequence length to capture global style attributes, e.g., timbre, prosody, and emotion; linguistic tokens $\mathbf{S}^{ling}$ to encode the content of the source utterance; and semantic tokens $\mathbf{S}^{sem}$ to represent the expressive target utterance and can be directly decoded into waveform when combined with speaker tokens.

The overall pipeline follows a single-stage autoregressive generation process:

$$\mathbf{X}_{src} \xrightarrow{\text{Tokenize}} (\mathbf{S}^{spk}_{src}, \mathbf{S}^{ling}_{src}) \xrightarrow{\text{Speech Translate}} (\mathbf{S}^{spk}_{src}, \mathbf{S}^{sem}_{tgt}) \xrightarrow{\text{Detokenize}} \mathbf{Y}_{tgt}.$$

This unified design allows UniSS to handle expressive S2ST without intermediate acoustic representations or cascaded systems, maintaining fidelity in both content and voice.

### 3.1.1 UNIFIED TEXT-SPEECH LANGUAGE MODEL

We build our generation model upon the pre-trained Qwen2.5-1.5B-Instruct (Yang et al., 2024), a strong LLM backbone. By expanding the model vocabulary to include the discrete speech tokens, UniSS treats speech and text uniformly as token sequences, allowing both modalities to be processed within the same transformer architecture.

The model takes a concatenated prompt of speaker tokens $\mathbf{S}^{spk}_{src}$ and linguistic tokens $\mathbf{S}^{ling}_{src}$ as input, and autoregressively generates the semantic token sequence $\mathbf{S}^{sem}_{tgt}$. The model is trained using a

Figure 2: Overview of UniSS, cross-modal CoT prompting, and 3-phase progressive training. Control Tokens instruct the task (S2ST Quality Mode in the figure), target language (Chinese in the figure), and output speed ratio compared with input (usually 1.0 means 1:1 duration matching).

standard next-token prediction objective:

$$\mathcal{L}_{\text{AR}} = -\sum_{t=1}^{|\tau_{out}|} \log P_\theta(\tau_{out,t}|\mathbf{P}, \tau_{out,<t}), \tag{1}$$

where $P_\theta(\cdot|\cdot)$ is the conditional probability of the next token given the context, as modeled by the LLM with parameters $\theta$. $\mathbf{P}$ is the input prompt containing $(\mathbf{S}_{src}^{spk}, \mathbf{S}_{src}^{ling})$ and $\tau_{out,<t}$ denotes the sequence of all previously generated tokens. $\tau_{out}$ represents the target sequence containing text tokens, semantic tokens, or both, depending on the specific task mode.

### 3.1.2 SPEECH TOKENIZER

To represent both the content and expressive characteristics of speech, UniSS adopts a triple-tokenizer strategy, transforming the waveform $\mathbf{W}$ into single-stream discrete token sequences:

$$(\mathbf{S}^{spk}, \mathbf{S}^{ling}, \mathbf{S}^{sem}) = \text{Tokenizer}(\mathbf{W}). \tag{2}$$

Here, $\mathbf{S}^{spk}$ and $\mathbf{S}^{sem}$ are produced by the **speaker tokenizer** and **semantic tokenizer**, respectively, both introduced in BiCodec (Wang et al., 2025). Specifically, the speaker tokenizer uses BiCodec's global encoder to extract speaker-related attributes, such as timbre, emotion, and prosody, into a fixed-length sequence of 32 tokens $\mathbf{S}^{spk}$. The semantic tokenizer captures other semantically relevant information from the waveform and encodes it into $\mathbf{S}^{sem}$ at 50 tokens per second.

Meanwhile, $\mathbf{S}^{ling}$ is generated by the **linguistic tokenizer**, which adopts the GLM-4 speech tokenizer (Zeng et al., 2024). It leverages a quantized Whisper encoder (Radford et al., 2023) to convert speech into a variable-length sequence of linguistic tokens at a rate of 12.5 tokens per second, enabling robust content understanding.

In the S2ST task, the source waveform $\mathbf{X_{src}}$ is represented by $(\mathbf{S}_{src}^{spk}, \mathbf{S}_{src}^{ling})$, while the target waveform $\mathbf{Y_{tgt}}$ is represented by $\mathbf{S}_{tgt}^{sem}$. This design is motivated by our preliminary experiments, which revealed that although BiCodec's semantic tokens are highly effective for waveform reconstruction, their self-supervised nature makes them suboptimal for content understanding. By separately representing speaker identity, linguistic content, and generation-oriented semantics, UniSS enables more accurate and controllable S2ST modeling.

### 3.1.3 SPEECH DETOKENIZER

We employ the BiCodec decoder to reconstruct the waveform $\mathbf{Y}_{tgt}$ from the semantic tokens $\mathbf{S}_{tgt}^{sem}$ generated by the LLM. The decoder is conditioned on the source speaker tokens $\mathbf{S}_{src}^{spk}$ to ensure voice and emotional style consistency. The decoding process is based on the simple concatenation of speaker and semantic tokens, represented as:

$$\mathbf{Y}_{tgt} = \text{Decoder}([\mathbf{S}_{src}^{spk}, \mathbf{S}_{tgt}^{sem}]). \tag{3}$$

This single-stage process directly reconstructs high-fidelity audio at a 16kHz sampling rate.

### 3.2 CROSS-MODAL CHAIN-OF-THOUGHT PROMPTING

Direct speech-to-speech translation is an inherently complex task. Inspired by the success of chain-of-thought (CoT) prompting in LLM, we introduce a cross-modal CoT prompting framework to decompose this task into a more manageable sequence of *listen*, *translate*, and *speak* steps. This approach is designed to effectively transfer LLM's powerful, pre-trained text translation capabilities

to the speech domain. Our framework provides two controllable prompts to trade off translation fidelity and efficiency.

UniSS is guided by an input prompt $\mathbf{P}$, which is a structured sequence of control and source tokens:

$$\mathbf{P} = [c_{task}, c_{lang}^{tgt}, c_{speed}, \mathbf{S}_{src}^{spk}, \mathbf{S}_{src}^{ling}],$$

where $c_{task}$, $c_{lang}^{tgt}$, and $c_{speed}$ are special tokens specifying the task mode, target language, and duration ratio between source and target speech, respectively. A special begin-of-translation token, BOT, signals the model to begin generation, producing an output sequence $\tau_{out}$ that is terminated by an end-of-decoding token, EOD.

**Quality Mode.** This mode follows the full CoT prompting path to maximize translation fidelity. The model first *listens* by generating the source transcription $\mathbf{T}_{src}$, then *translates* it into the text in the target language $\mathbf{T}_{tgt}$, and finally *speaks* by generating target semantic tokens $\mathbf{S}_{tgt}^{sem}$. Prompted with $c_{task} = $ Quality Mode, this explicit chain allows the model to leverage its robust text translation abilities, formally represented as:

$$\tau_{out} = [\mathbf{T}_{src}, \mathbf{T}_{tgt}, \mathbf{S}_{tgt}^{sem}].$$

**Performance Mode.** This mode compresses the CoT path for faster inference. It skips the transcription step and directly generates the target text translation $\mathbf{T}_{tgt}$ followed by the corresponding semantic tokens $\mathbf{S}_{tgt}^{sem}$. Prompted with $c_{task} = $ Performance Mode, the output becomes:

$$\tau_{out} = [\mathbf{T}_{tgt}, \mathbf{S}_{tgt}^{sem}].$$

### 3.3 PROGRESSIVE TRAINING STRATEGY

We adopt a three-phase progressive training strategy to optimize the model, aiming to gradually build cross-modal capabilities while mitigating catastrophic forgetting of the LLM's foundational text translation abilities.

**Phase 1: Speech-Text Alignment.** We begin by adapting the pre-trained LLM to the speech modality through a multi-task learning stage involving four foundational tasks: ASR, TTS, Speech-to-Text Translation (S2TT), and MT. The ASR, TTS, and S2TT tasks align speech and text, while MT preserves the model's foundational translation capabilities. Each task is defined by a specific prompt structure and target output sequence:

- ASR: $\mathbf{P} = [c_{asr}, c_{lang}^{src}, \mathbf{S}_{src}^{spk}, \mathbf{S}_{src}^{ling}]$, $\tau_{out} = \mathbf{T}_{src}$.

- S2TT: $\mathbf{P} = [c_{s2tt}, c_{lang}^{tgt}, \mathbf{S}_{src}^{spk}, \mathbf{S}_{src}^{ling}]$, $\tau_{out} = \mathbf{T}_{tgt}$.

- TTS: $\mathbf{P} = [c_{tts}, c_{lang}^{src}, \mathbf{S}_{src}^{spk}, \mathbf{T}_{src}]$, $\tau_{out} = \mathbf{S}_{src}^{sem}$.

- MT: $\mathbf{P} = [c_{mt}, c_{lang}^{tgt}, \mathbf{T}_{src}]$, $\tau_{out} = \mathbf{T}_{tgt}$.

Here $c_{lang}^{src}$ denotes the source language, and $\mathbf{S}_{src}^{sem}$ denotes the semantic tokens from source speech. This phase endows the model with robust speech understanding and generation while prioritizing the preservation of text translation capabilities.

**Phase 2: S2ST with CoT.** In the second phase, we introduce the core S2ST task. The model is trained to generate outputs using the CoT prompting formats described in Section 3.2, as well as a simplified direct generation mode that bypasses intermediate text outputs:

$$\mathbf{P} = [c_{s2st}, c_{lang}^{tgt}, c_{speed}, \mathbf{S}_{src}^{spk}, \mathbf{S}_{src}^{ling}], \tau_{out} = \mathbf{S}_{tgt}^{sem}.$$

This phase utilizes the alignment from Phase 1 and transfers the LLM's text translation capabilities to the speech domain.

**Phase 3: Refinement.** In the final phase, we fine-tune the model on the full S2ST task using both CoT prompting modes. This phase uses an annealed learning rate to stabilize the learned CoT patterns and optimize the final translation performance.

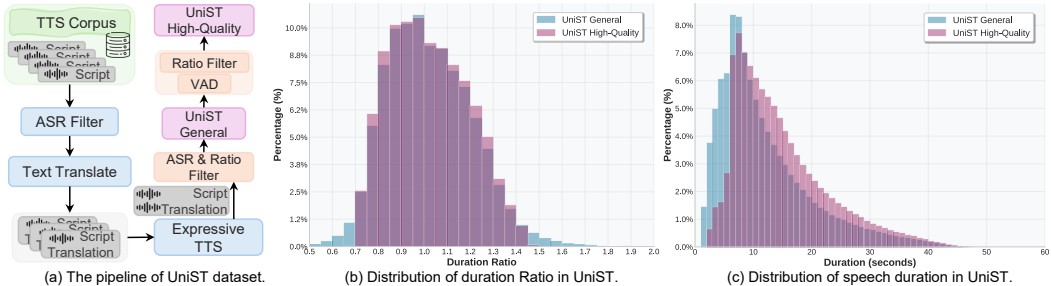

(a) The pipeline of UniST dataset.    (b) Distribution of duration Ratio in UniST.    (c) Distribution of speech duration in UniST.

Figure 3: Overview of the UniST data construction pipeline and analysis of duration ratio and speech duration distributions.

## 4 THE UNIST DATASET

Training end-to-end expressive S2ST models faces a significant bottleneck due to the scarcity of large-scale, parallel data that preserves speaker characteristics. To address this limitation, we design a scalable data synthesis pipeline. The pipeline generates UniST, our large-scale Chinese-English expressive S2ST dataset. Figure 3(a) illustrates the data construction process.

**Data Sources and Cleaning.** Our pipeline begins with large-scale TTS corpora containing paired speech and transcription $(\mathbf{X}_{src}, \mathbf{T}_{src})$, combined from several Chinese and English public datasets (details in Appendix E.1). Following VoxBox (Wang et al., 2025), we first clean the corpus using Paraformer (Gao et al., 2022) to re-recognize speech and compute Word Error Rate (WER) between re-recognized and original transcriptions, discarding samples with WER > 0.05.

**Translation and Target Speech Synthesis.** After source cleaning, $\mathbf{T}_{src}$ is translated by Qwen2.5-72B-Instruct (Yang et al., 2024) to produce target text $\mathbf{T}_{tgt}$ in another language. Prompts used in translation are shown in Appendix E.2. We then apply an expressive TTS model, SparkTTS (Wang et al., 2025), to synthesize the target speech $\mathbf{Y}_{tgt}$ from $\mathbf{T}_{tgt}$, conditioned by $\mathbf{X}_{src}$ to preserve the source speaker's voice. To enable fine-grained speed control, we calculate the duration ratio between source and target speech and discretize it into speed tokens $\mathbf{c}_{speed}$ with 0.1 intervals. This creates complete parallel samples $(\mathbf{X}_{src}, \mathbf{T}_{src}, \mathbf{T}_{tgt}, \mathbf{Y}_{tgt}, \mathbf{c}_{speed})$.

**Quality Filtering and Dataset Variants.** The synthesized data undergoes final quality filtering. We apply ASR to $\mathbf{Y}_{tgt}$ and discard samples with WER greater than 0.01 against $\mathbf{T}_{tgt}$. We also apply a duration ratio filter to keep synthesized speech with duration within [0.5, 2.0] times of the source speech. The filtered data forms our **UniST General** dataset (44.8k hours). Our refined **UniST High-Quality** dataset (19.8k hours) is created by applying additional Voice Activity Detection (Team, 2024) to remove silence at the beginning and end of speech segments, and a stricter duration ratio filter of [0.7,1.5].

As shown in Figure 3(b,c), UniST General provides greater data diversity and generalization potential due to its larger range of duration ratios and speech lengths. UniST High-Quality focuses on temporal consistency, making it ideal for the final refinement phase of our progressive training. In Appendix E.3, we show that the UniST dataset also preserves emotional style in synthesis.

## 5 EXPERIMENTS AND RESULTS

### 5.1 TRAINING SETUP

UniSS is trained in three progressive phases, as introduced in Section 3.3. Phase 1 establishes text-speech alignment using 77.1k hours of speech data and MT text data from WMT17 (Bojar et al., 2017). Phase 2 introduces CoT prompting with our UniST General dataset, mixed with Phase 1 data at a 2:1 ratio. Phase 3 fine-tunes the model exclusively on the UniST High-Quality dataset.

Training employs the AdamW optimizer (Loshchilov & Hutter, 2019) with a 2.3M-token batch size, weight decay of 0.1, and momentum parameters (0.9, 0.95). All audio is resampled to 16 kHz, and the LLM vocabulary is expanded to 180,407 to include speech and control tokens. Learning rates progress from 8e-4 in Phase 1 to 2e-4 in Phase 2, and finally anneal from 5e-5 to 5e-6 in Phase 3. The model is trained on 16 NVIDIA H800 80G GPUs using the Megatron-LM Framework (Shoeybi et al., 2019) for efficient large-model training. Complete training details and hyperparameters are provided in Appendix B.1.

## 5.2 EVALUATION SETUP

### 5.2.1 DATASETS

For our primary S2ST evaluation, we use the Chinese (ZH) and English (EN) test sets from CVSS-T (Jia et al., 2022b), which contain 4,897 utterance pairs (8.2 hours for Chinese and 6.3 hours for English). For emotion preservation evaluation, we randomly sample 300 utterances from each emotional speech dataset (ESD (Zhou et al., 2021) and CREMA-D (Cao et al., 2014)), covering a selected set of 4 emotions (happy, sad, angry, neutral). We evaluate both English-to-Chinese (EN-ZH) and Chinese-to-English (ZH-EN) translation directions.

### 5.2.2 METRICS

We employ comprehensive objective and subjective metrics to evaluate S2ST performance. For translation fidelity, we measure BLEU scores (Papineni et al., 2002) on both ASR-transcribed speech outputs (Speech-BLEU) and intermediate text (Text-BLEU). Prosody preservation is assessed through prosody similarity (A.PCP) (Andrews et al., 2022). Duration consistency is evaluated using Speech Length Compliance (SLC) within 0.2 and 0.4 ratio tolerance (Wu et al., 2023). Speech quality is measured using UTMOS (Saeki et al., 2022). Additionally, we conduct a subjective Mean Opinion Score (MOS) evaluation where bilingual speakers rate emotion similarity, speaker similarity, and naturalness on a 5-point scale. Detailed descriptions of all evaluation metrics and implementation procedures are provided in Appendix C.

### 5.2.3 BASELINES

We compare UniSS against a strong and diverse set of state-of-the-art (SOTA) systems. Details of baseline implementation are introduced in Appendix B.2.

- **Cascaded Systems.** To establish an upper bound for modular approaches, we construct three cascaded baselines using top-performing open-source models: a **3-Stage-MT** pipeline of Whisper-large-v3 (ASR), NLLB-200-distilled (MT) (Koishekenov et al., 2023), and CosyVoice 2 (TTS) (Du et al., 2024); a **3-Stage-LLM** pipeline of Whisper-large-v3 (ASR), Qwen2.5-1.5b-Instruct (MT), and CosyVoice 2 (TTS); and a **2-Stage** pipeline of SeamlessM4T-v2-Large (S2TT) and CosyVoice 2 (TTS).

- **Multimodal Large Language Models (MLLMs).** To benchmark against the latest generation of general-purpose models, we include two leading MLLMs: **GPT-4o** (Hurst et al., 2024), an enterprise-level model from OpenAI with strong speech-to-speech capability; and **Qwen2.5-Omni** (Xu et al., 2025), a powerful open-source MLLM building on large-scale audio–language pretraining, enabling both speech understanding and synthesis.

- **End-to-End S2ST Systems.** We compare against dedicated S2ST models that represent the current open-source SOTA: **SeamlessM4T** (Medium and Large V2, denoted Seamless-M and Seamless-L), and its expressive variant **SeamlessExpressive** (Seamless-Ex). For subjective metrics, we also evaluate **Seed LiveInterpret 2.0** (Cheng et al., 2025), denoted as Seed Live, an enterprise-level S2ST system with duplex speech understanding and generation abilities.

### 5.2.4 INFERENCE CONFIGURATION

For UniSS experiments, we deploy vLLM (Kwon et al., 2023) to support inference. We set a decoding temperature of 0.7, top-k of -1, top-p of 0.8, and a repetition penalty of 1.1. We report results for both Performance mode, denoted as **UniSS (P)**, and Quality mode, denoted as **UniSS (Q)**.

## 5.3 OBJECTIVE PERFORMANCE

Table 1 presents the main comparison results on the CVSS-T dataset. The results clearly demonstrate that UniSS establishes a new SOTA in translation fidelity while maintaining voice characteristics, prosody, duration consistency, and speech quality, validating our core design principles. Additional results on other datasets and tasks are provided in Appendix D.

**Translation Fidelity.** UniSS achieves state-of-the-art translation fidelity on both EN-ZH and ZH-EN directions. The UniSS (Q) variant achieves a Speech-BLEU of 32.20 on EN-ZH and 24.28 on ZH-EN, substantially outperforming all prior end-to-end and cascaded baselines. The efficient UniSS (P) also delivers strong results, surpassing most existing systems. Notably, in terms of intermediate text metrics, UniSS models perform on par with or better than larger multimodal LLM-based approaches.

Table 1: Main comparison results on the CVSS-T dataset. Results are presented as EN-ZH | ZH-EN. Higher scores indicate better performance. '-' denotes unavailable results. Best scores are in **bold** and second-best scores are underlined.

| Category | Model | #Size | Speech-BLEU | Text-BLEU | A.PCP | SLC 0.2 | SLC 0.4 | UTMOS |
|---|---|---|---|---|---|---|---|---|
| GT | CVSS-T | - | - | - | 2.37 | 0.32 | 0.72 | 2.27 \| 3.56 |
| Cascaded | 3-Stage-MT | 2.6B | 25.02 \| 16.62 | 25.80 \| 17.08 | 2.80 \| 2.85 | 0.56 \| 0.54 | 0.82 \| 0.88 | 3.76 \| 3.50 |
| | 3-Stage-LLM | 3.5B | 26.74 \| 17.69 | 28.36 \| 18.51 | 2.84 \| 2.82 | 0.60 \| 0.48 | 0.87 \| 0.84 | 3.76 \| 3.48 |
| | 2-Stage | 2.8B | 26.94 \| 20.86 | 27.38 \| 22.20 | **2.87** \| 2.64 | 0.67 \| 0.52 | 0.93 \| 0.70 | **3.79** \| 3.48 |
| MLLM | GPT-4o | - | 31.64 \| 19.27 | - \| - | 2.66 \| 2.58 | 0.47 \| 0.37 | 0.71 \| 0.61 | 3.46 \| 4.18 |
| | Qwen2.5-O | 7B | 7.10 \| 22.66 | **34.85** \| 24.39 | 1.90 \| 1.92 | 0.31 \| 0.35 | 0.57 \| 0.61 | 3.23 \| **4.30** |
| S2ST | Seamless-M | 1.2B | 14.53 \| 14.36 | 24.80 \| 18.44 | 2.34 \| 2.29 | 0.54 \| 0.22 | 0.82 \| 0.45 | 2.73 \| 3.59 |
| | Seamless-L | 2.3B | 25.05 \| 17.67 | 27.61 \| 21.95 | 2.41 \| 2.15 | 0.67 \| 0.36 | 0.95 \| 0.62 | 2.69 \| 4.04 |
| | Seamless-Ex | 1.7B | 24.45 \| 15.84 | 26.59 \| 16.74 | 2.83 \| **2.87** | 0.68 \| 0.52 | 0.94 \| 0.77 | 2.46 \| 2.90 |
| | UniSS (P) | 1.5B | 30.28 \| 23.61 | 30.93 \| 24.45 | 2.73 \| 2.75 | **0.98** \| 0.84 | **0.99** \| **0.97** | 3.77 \| 3.86 |
| | UniSS (Q) | 1.5B | **32.20** \| **24.28** | 32.95 \| **26.28** | 2.71 \| 2.74 | **0.98** \| **0.87** | **0.99** \| **0.97** | 3.76 \| 3.86 |

**Prosody Preservation.** In terms of prosody, UniSS achieves competitive performance. UniSS (P) variant achieves the second-highest A.PCP score (2.73 and 2.75), closely following Seamless-Ex, which incorporates a dedicated prosody encoder. The performance gap is marginal at only 0.10 (EN-ZH) and 0.12 (ZH-EN), highlighting the efficacy of UniSS in preserving prosodic patterns without specialized modules.

**Duration Consistency.** UniSS demonstrates superior duration consistency. UniSS (Q) achieves near-optimal SLC 0.2 scores on EN-ZH and the best performance on ZH-EN, improving over the previous best end-to-end system (Seamless-Ex) by 44% and 67%. On the more relaxed SLC 0.4 metric, while competing systems achieve scores above 0.90, both UniSS variants deliver near-perfect performance with scores of 0.99 (EN-ZH) and 0.97 (ZH-EN).

**Speech Quality.** UniSS achieves state-of-the-art speech quality among S2ST models. In the EN-ZH direction, UniSS (P) achieves a competitive UTMOS score of 3.77, matching cascaded models (3.76 and 3.79) while surpassing all end-to-end models. In the ZH-EN direction, UniSS surpasses 3-stage systems with a score of 3.86 compared to 3.50.

## 5.4 Subjective Results

In addition to the objective metrics, we conduct comprehensive subjective assessments on emotion preservation, voice preservation, and speech naturalness. The results are shown in Table 2.

**Emotion Preservation.** UniSS (Q) demonstrates strong capability in preserving emotions, achieving a MOS of 4.51. This represents a substantial 27% improvement over the expressive S2ST baseline Seamless-Ex (3.56) and surpasses the 3-Stage-MT cascaded system (4.48). Notably, UniSS (Q) also approaches Seed Live (4.56), indicating its ability in capturing emotional nuance.

**Speaker Preservation.** UniSS effectively retains speaker voice characteristics without requiring an additional NAR stage-2 model. For speaker similarity, UniSS (Q) achieves a score of 4.42, outperforming all other models. This represents a 0.07 improvement over the 2-stage system (4.35), which deploys a carefully designed TTS model. These results underscore the ability of UniSS to maintain voice characteristics in an end-to-end fashion.

**Speech Naturalness.** The naturalness of speech from UniSS is also favorably rated. UniSS (Q) achieves a naturalness MOS of 4.45, surpassing Seamless-Ex (3.10) and the 3-stage cascaded baseline (4.31) that incorporates a specialized TTS model. This demonstrates UniSS's ability to generate high-quality, natural-sounding speech without dedicated text-to-speech components.

## 5.5 Inference Speed and Quality Trade-off

Our framework provides flexible control over the quality-efficiency trade-off through its different CoT prompting modes. We evaluate inference speed on the AR language model component using the Transformers library (Wolf et al., 2020), using 400 utterances (200 per direction) from CVSS-T on a single H800 GPU without batching.

Table 3 shows the Performance mode achieves a $1.07\times$ speedup over Quality mode with only a 1.84 point reduction in Speech-BLEU, demonstrating a favorable speed-quality trade-off. Furthermore, we trained **UniSS-Small** based on Qwen2.5-0.5B-Instruct, achieving significant computational sav-

Table 2: Subjective MOS evaluation on the expressive emotion dataset. *Seed Live is closed source. Best scores are in **bold** and second-best scores are underlined.

| Model | Emo Sim.↑ | Spk Sim.↑ | Naturalness↑ |
|---|---|---|---|
| 3-Stage-MT | 4.48 | 4.33 | 4.31 |
| 2-Stage | 4.27 | 4.35 | 4.27 |
| Seed Live* | **4.56** | 4.19 | **4.69** |
| Qwen2.5-O | 2.84 | N/A | 4.47 |
| Seamless-Ex | 3.56 | 2.94 | 3.10 |
| UniSS (Q) | 4.51 | **4.42** | 4.45 |

Table 3: Inference speed of AR language model and Speech-BLEU comparison. Speech-BLEU is the average of EN-ZH and ZH-EN. Time is the total inference time on 400 utterances without batching inference. Best results are in **bold**.

| Model | #Size | Speech-BLEU↑ | Time(s)↓ | Speedup↑ |
|---|---|---|---|---|
| UniSS (Q) | 1.5B | **33.66** | 1521.52 | 1.00× |
| UniSS (P) | 1.5B | 31.82 | 1426.54 | 1.07× |
| UniSS-Small (Q) | 0.5B | 28.17 | 1339.24 | 1.14× |
| UniSS-Small (P) | 0.5B | 25.68 | **1212.65** | **1.25×** |

ings. UniSS-Small (P) delivers 1.25× speedup with competitive translation fidelity (25.68 Speech-BLEU), making it suitable for resource-constrained deployment scenarios while maintaining the advantages of our unified architecture.

## 5.6 ABLATION STUDIES

We conduct ablation studies to validate the effect of our design. Table 4 presents the detailed results comparing variants against a base model trained through Phases 1 and 2. The results in type Base and Train are evaluated in the Performance mode.

### 5.6.1 IMPACT OF PROGRESSIVE TRAINING

Our three-phase progressive training significantly impacts performance. Phase 3 refinement (**w/ Phase 3**) contributes improvements of +0.90 and +2.06 Speech-BLEU points, validating the importance of high-quality data fine-tuning for final optimization. Removing the initial Phase 1 alignment (**UniST only**) causes severe performance degradation of -7.18 and -10.15 points. This drop demonstrates that Phase 1 text-speech alignment is essential for subsequent S2ST learning.

Table 4: Ablation study on UniSS components evaluated on CVSS-T test set. w/ and w/o denote 'with' and 'without', respectively. Best scores are in **bold**.

| Type | Variant | Speech-BLEU↑ | | Δ | |
|---|---|---|---|---|---|
| | | EN-ZH | ZH-EN | EN-ZH | ZH-EN |
| Base | Phase 1+2 | 29.38 | 21.55 | - | - |
| Train | w/ Phase 3 | **30.28** | **23.61** | +0.90 | +2.06 |
| | UniST only | 22.20 | 11.40 | -7.18 | -10.15 |
| | w/o GLM | 14.37 | 12.82 | -15.01 | -8.73 |
| Infer | Direct S2ST | 14.44 | 7.15 | -14.94 | -14.40 |

### 5.6.2 IMPACT OF LINGUISTIC TOKENIZER

Replacing our content-focused GLM-4 linguistic tokenizer with BiCodec's self-supervised semantic tokens (**w/o GLM**) leads to significant performance degradation of -15.01 and -8.73 Speech-BLEU points. This drop reveals that while BiCodec's semantic tokens excel at speech generation tasks, their self-supervised nature limits their effectiveness for content understanding in the S2ST task.

### 5.6.3 IMPACT OF CHAIN-OF-THOUGHT FRAMEWORK

Removing the intermediate text generation and performing direct speech-to-speech translation (**Direct S2ST**) results in a severe performance degradation of -14.94 and -14.40 Speech-BLEU points in the inference. This demonstrates that our cross-modal CoT prompting enables the transfer of textual translation expertise to the speech domain and improves translation fidelity.

## 6 CONCLUSION, LIMITATIONS AND FUTURE WORKS

This paper introduces UniSS, a unified single-stage expressive speech-to-speech translation framework that preserves voice and emotional style. Our approach eliminates architectural complexity through well-designed speech semantic and style modeling, enabling seamless integration with pretrained LLMs. By transferring LLMs' text translation capabilities to speech via cross-modal CoT prompting, UniSS surpasses previous S2ST systems across multiple dimensions: translation fidelity, speech naturalness, and preservation of voice, emotion, and duration consistency. Beyond the model architecture, we design a reproducible pipeline to create expressive S2ST datasets and construct UniST, a large-scale Chinese-English expressive S2ST dataset. Our work demonstrates a simple and effective approach for building the next generation of expressive S2ST systems.

**Limitations and Future Works.** Current limitations point to several research directions. (1) Language Support: We trained UniSS only on Chinese and English due to resource limitations. The data construction pipeline and training framework can be easily extended to multilingual scenarios, which represents our immediate next step. (2) Speech Tokenizer: While the linguistic, speaker, and semantic tokenizers perform well in UniSS, they originate from two different audio tokenizers, causing vocabulary size expansion. Future work will focus on training a unified tokenizer to merge these components and reduce vocabulary size.

## 7 ACKNOWLEDGEMENTS

The research was supported by Early Career Scheme (ECS-HKUST22201322), Theme-based Research Scheme (T45-205/21-N) from Hong Kong RGC, and Generative AI Research and Development Centre from InnoHK.

## ETHICS STATEMENT

We acknowledge that our work on speech-to-speech translation, which preserves a speaker's vocal identity, has the potential for dual use. Our primary goal is to foster more natural and personal cross-lingual communication, benefiting applications like international conferencing and film dubbing. However, the technology could be misused by malicious actors. Potential risks include the creation of audio deepfakes for disinformation campaigns, impersonation for fraud, or targeted harassment.

## REPRODUCIBILITY STATEMENT

We ensure that both the methods and the dataset pipeline are reproducible. Training details are provided in Appendix B.1. All raw data originates from open-source datasets, with details provided in Appendix E.

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

## A EXAMPLES

We present real examples from different models here, with speech converted to text using ASR for display purposes. As shown in Figure A1, UniSS achieves competitive BLEU scores on challenging sentences where GPT-4o struggles with translation quality. Note Seamless-L and GPT-4o cannot preserve source voice and emotional style.

| Source | GT | Models | ASR Infer | BLEU |
|---|---|---|---|---|
| Civil legal act is the corresponding concept between China and Germany. | 中国与德国对应的概念是民事法律行为。 | UniSS | 民事法律行为是中国与德国之间的相对概念。 | **49.5** |
| | | Seamless-Ex | 民事法律行为是中国和德国之间的相应概念。 | 22.9 |
| | | Seamless-L | 民事法律行为是中国和德国之间的相应概念。 | 22.9 |
| | | GPT-4o | 中国和德国之间的对应概念是民事法。 | 38.6 |
| When nurses enter the workplace these badges are used to identify their alma mater. | 当护理人员进入职场后，这些护理徽章就用以识别其母校。 | UniSS | 当护士进入工作场所时这些徽章用于识别他们的母校。 | **8.0** |
| | | Seamless-Ex | 当护士进入工作场所时这些规章被用来识别他们的母校。 | 7.5 |
| | | Seamless-L | 当护士进入工作场所时这些安克装备用来识别他们的母校。 | 7.0 |
| | | GPT-4o | 当护士进入工作场所时这些徽章被用来识别他们的母校。 | **8.0** |
| 这些夫妇可能会为他们的孩子制定收养计划。 | These couples may choose to make an adoption plan for their baby. | UniSS | These couples may develop an adoption plan for their children. | **43.0** |
| | | Seamless-Ex | These couples are likely to draw up adoption plans for their children. | 10.6 |
| | | Seamless-L | These couples may make adoption plans for their children. | 17.4 |
| | | GPT-4o | These couples may make adoption plans for their children. | 17.4 |
| 他还在金星的红外光谱中发现了强烈的二氧化碳谱线。 | He also found a strong carbon dioxide spectral line in the infrared spectrum of the venus. | UniSS | It also found strong carbon dioxide spectral lines in the infrared spectrum of Venus. | **42.9** |
| | | Seamless-Ex | It has also detected strong Codwell lines in the carefully mapped X-rays. | 5.8 |
| | | Seamless-L | He also found a strong carbon dioxide light source in the red light pool. | 41.5 |
| | | GPT-4o | It has detected strong carbon dioxide spectral lines in the infrared spectra. | 22.0 |

Figure A1: Comparison of speech translation outputs from different models. Examples show source speech (converted to text via ASR), ground truth translations, and model outputs with corresponding BLEU scores. UniSS demonstrates superior performance on challenging sentences where other models struggle. Best scores are in **bold** and second-best scores are underlined.

## B IMPLEMENTATION DETAILS

### B.1 UNISS

Table B1: UniSS training configuration showing datasets, hyperparameters, and training settings for each phase.

| Phase | Dataset | Training Tokens | Epochs | Learning Rate | Warm-up | Batch Size |
|---|---|---|---|---|---|---|
| **Phase 1** | 77.1k hours speech data 2.3B MT tokens | ~32B per epoch | 3 | 8e-4 (constant) | 1 epoch | 2.3M |
| **Phase 2** | UniST General dataset Mixed with Phase 1 data (2:1) | ~55B total | 1 | 2e-4 (constant) | 5% epoch | 2.3M |
| **Phase 3** | UniST High-quality | ~10B total | 0.9/1 | 5e-5 → 5e-6 (cosine) | None | 2.3M |

#### B.1.1 TRAINING PROTOCOL

As shown in Table B1, the UniSS is trained in three phases progressively. All audio is resampled to 16 kHz. The LLM vocabulary is expanded to 180,407 to include speech and control tokens. We use the AdamW optimizer with a weight decay of 0.1 and momentum parameters (0.9, 0.95). The batch size is fixed at 2.3M tokens for all phases.

**Phase 1: Text-Speech Alignment.** The model is trained on a mixture of ASR, TTS, S2TT, and MT tasks. This phase utilizes 77.1k hours of speech data and 2.3B translation tokens from WMT17, totaling approximately 32B tokens per epoch. The model is trained for 3 epochs with a constant learning rate of 8e-4 and a 1-epoch warm-up.

**Phase 2: Chain-of-Thought Prompting.** We introduce CoT prompting using our UniST General dataset. The model learns three generation paths: Performance mode, Quality mode, and direct

S2ST data without intermediate text. New data is mixed with Phase 1 data at a 2:1 ratio, totaling approximately 55B tokens. We train for one epoch with a constant learning rate of 2e-4 and a 5% warm-up.

**Phase 3: Annealing.** We fine-tune the model exclusively on the UniST High-quality dataset. The learning rate follows a cosine schedule, gradually annealing from 5e-5 to 5e-6 over 1 epoch without warm-up. This phase consists of approximately 10B training tokens. During this stage, the training loss reaches its minimum around 0.9 epochs, and we select the checkpoint at this point as our final model weights.

### B.1.2 IMPLEMENTATION DETAILS

Our language model is trained on 16 NVIDIA H800 80G GPUs. We utilize the Megatron-LM Framework for efficient large-model training. Because audio duration distribution is uneven, padding would waste significant computational resources. We use the sequence packing (Krell et al., 2022) technique to concatenate multiple samples into a single 18k token long sequence. We use a global batch size of 128. Completing all three phases of training takes approximately 6 days.

### B.2 BASELINES

Table B2: Overview of baseline models used in experiments.

| Pipeline | Stage | Model Name | Model Source | Size |
|---|---|---|---|---|
| Three-stage | ASR | Whisper-large-v3 | openai/whisper-large-v3 | Large (∼1.5B) |
| | MT | NLLB-200-distilled-600M | facebook/nllb-200-distilled-600M | 600M |
| | TTS | CosyVoice 2 | FunAudioLLM/CosyVoice2-0.5B | 0.5B |
| Two-stage | Speech Translation | SeamlessM4T-v2-Large | facebook/seamless-m4t-v2-large | Large (∼2.3B) |
| | TTS | CosyVoice 2 | FunAudioLLM/CosyVoice2-0.5B | 0.5B |
| Multimodal LLMs | Speech-to-Text | Qwen2-audio | Qwen/Qwen2-audio | 7B |
| | Speech-to-Speech | Qwen2.5 Omni | Qwen/Qwen2.5-Omni | 7B |
| | Speech-to-Speech | GPT-4o-Audio-Preview | OpenAI Official API | Not disclosed |
| End-to-End S2ST | S2ST | SeamlessM4T Medium | facebook/seamless-m4t-medium | Medium (∼1.2B) |
| | S2ST | SeamlessM4T Large-v2 | facebook/seamless-m4t-large-v2 | Large (∼2.3B) |
| | S2ST | SeamlessExpressive | facebook/seamless-expressive | ∼1.7B |

In this section, we provide detailed descriptions of all baseline models evaluated against our proposed UniSS framework. Table B2 summarizes the baselines, model versions, and parameter sizes used in our experiments. All baseline models are evaluated using their default inference parameters without any additional tuning or modifications. Evaluation is conducted directly on the CVSS-T and FLEURS test sets without extra data preprocessing, prompts, or output post-processing. We present the implementation specifics of each baseline in the following subsections.

### B.2.1 CASCADED SYSTEMS

We implement two cascaded baselines: a conventional three-stage baseline (ASR → MT → TTS) and a two-stage baseline (S2TT → TTS). We select state-of-the-art, widely adopted models at each stage to represent strong and practically relevant baselines.

**Three-Stage Baseline (Whisper + NLLB + CosyVoice).** This baseline consists of three sequential modules:

- **Automatic Speech Recognition (ASR)**: We select Whisper-large-v3, a transformer-based encoder-decoder model trained on large-scale multilingual speech corpora, for its strong multilingual transcription performance and wide adoption in the community.

- **Machine Translation (MT)**: We use Meta's NLLB-200-distilled-600M model, a multilingual transformer trained on parallel corpora covering 200 languages. We select this model for its balance between translation quality and computational efficiency.

- **Text-to-Speech (TTS)**: We employ CosyVoice 2 (0.5B), a transformer-based expressive TTS model designed for high-quality multilingual speech synthesis. Its ability to generate natural and expressive audio makes it suitable for our evaluation.

**Two-Stage Baseline (SeamlessM4T-v2-Large + CosyVoice).** The two-stage pipeline simplifies the cascaded approach by combining speech recognition and translation into one direct Speech-to-Text Translation (S2TT) module:

- **Speech-to-Text Translation (S2TT)**: We select SeamlessM4T-v2-Large, a multilingual transformer-based model that jointly performs ASR and MT. Its end-to-end design reduces error propagation across stages and offers strong performance in speech translation.

- **Text-to-Speech (TTS)**: We generate synthesized speech using the same CosyVoice 2 model described in the three-stage pipeline.

In our experiments, we observe several limitations associated with cascaded baselines. First, the inference process is noticeably slower due to the sequential execution of multiple large models, making the pipeline less suitable for real-time applications. Second, we find that errors introduced in earlier stages, particularly in ASR, tend to propagate downstream, often leading to degraded translation quality or unnatural speech synthesis. These limitations highlight the challenges of building efficient and robust cascaded speech translation systems.

### B.2.2 MULTIMODAL LARGE LANGUAGE MODELS (MLLM)

We employ three advanced multimodal large language models as baselines: Qwen2-audio, Qwen2.5-Omni, and GPT-4o-audio. These models are capable of handling speech input and producing either textual or speech output, depending on the task.

**Qwen2.5-Omni (Speech-to-Speech).** Qwen2.5-Omni is selected for its ability to perform fully end-to-end speech-to-speech translation without requiring intermediate text generation, making it a promising multimodal LLM baseline for direct S2ST tasks. In our experiments, we find that the model's performance is highly sensitive to the prompt format. When using a simple default instruction, the model sometimes appends assistant-like phrases (e.g., "Do you need anything else?") to the translated speech, likely due to its conversational fine-tuning.

```
conversation = [
{
"role": "user",
"content": [
    {
        "type": "text",
        "text": "Translate the following Chinese speech into
        ↪  English."
    },
    {
        "type": "audio",
        "audio": input_path
    }
],
},
]
```

To mitigate this, we also experimented with more explicit and constrained prompts, such as:

```
conversation = [
{
"role": "user",
"content": [
{
    "type": "text",
    "text": "You are an expert translator specializing in
    ↪   Chinese-English interpretation. Your task is to listen to
    ↪   the following Chinese speech and translate it accurately and
    ↪   fluently into spoken English. Your response should: -
    ↪   Preserve the original meaning and tone of the speaker. - Use
    ↪   natural and professional English phrasing. - Output only the
    ↪   translated speech without adding any commentary or
    ↪   assistant-like phrases. - Maintain appropriate speech rhythm
    ↪   and avoid stretching or pausing unnaturally. Please begin
    ↪   the translation after receiving the audio input."
},
{
    "type": "audio",
    "audio": input_path
}
],
},
]
```

However, we find that such detailed prompts often lead to degraded audio output quality. Specifically, in our experiments, the model occasionally produces abnormal acoustic artifacts, such as elongating a single word unnaturally or producing disfluent prosody. This suggests that Qwen2.5-Omni's speech generation may be overly sensitive to instruction format, and that more verbose prompts can negatively impact generation fluency despite offering better control over semantic behavior.

**Qwen2-audio (Speech-to-Text).** Qwen2-audio (Chu et al., 2024) is used for direct speech-to-text translation. Considering its strong performance on multilingual speech comprehension tasks and its ability to follow structured prompts, we adopt it for S2TT evaluation. Prompts are designed to explicitly guide the model to translate input speech into target-language text. For example:

```
<|audio_bos|><|AUDIO|><|audio_eos|> Translate Chinese into English.
```

All other parameters are kept at their default values. However, using Qwen2-audio requires additional input-output processing. Specifically, all input audio is resampled to 16 kHz and converted to mono channel to meet the model's input requirements. For output handling, we filter the returned results to remove prompt-related metadata and retain only the translated text for evaluation purposes.

**GPT-4o.** For S2ST evaluation, we employ GPT-4o-audio-preview, which supports multimodal speech translation tasks with direct speech-to-speech generation capabilities. Given the instability of Qwen2.5-Omni under complex prompt instructions observed in preliminary experiments, we select GPT-4o-audio-preview as a more robust alternative. Our experiments confirm that GPT-4o-audio-preview demonstrates superior stability and consistently produces higher-quality outputs compared to previous models.

We guide the model using simple structured prompts such as:

```
Please translate this English audio into Chinese. Just state the
↪   translation itself without any additional explanation or polite
↪   words.
```

Although we occasionally encounter conversational artifacts, such as appended phrases like "Do you need help with anything else?" or generic fallback responses like "I'm sorry, I can't process this audio", the overall translation fluency and speech quality are significantly improved compared to

earlier baselines. However, GPT-4o is not open-sourced, and using its API for large-scale translation incurs considerable cost, which poses a practical constraint in real-world deployment scenarios.

### B.2.3  END-TO-END SPEECH-TO-SPEECH MODELS

For direct end-to-end baselines, we evaluate several state-of-the-art S2ST systems that translate input audio directly into target-language speech, bypassing intermediate textual representations. Specifically, we include SeamlessM4T Medium, SeamlessM4T Large-v2, and SeamlessExpressive in our evaluation.

**SeamlessM4T Medium and Large-v2.**  SeamlessM4T Medium and Large-v2 are multilingual speech translation models that support speech-to-speech translation without explicit intermediate transcription or synthesis stages. These models adopt a unified encoder-decoder architecture and are designed to support both low-resource and high-resource language pairs. In our experiments, we use the official inference pipelines provided by Meta with default decoding parameters. No additional prompt tuning, input preprocessing, or output filtering is applied.

**SeamlessExpressive.**  SeamlessExpressive extends SeamlessM4T by introducing expressiveness-aware modeling, enabling more natural and emotionally rich speech synthesis in the target language. The model conditions generation on prosodic and expressive cues from the source speech and is particularly well-suited for conversational and affective scenarios. We evaluate SeamlessExpressive using its official inference scripts and default hyperparameters.

**Seed LiveInterpret 2.0.**  Seed LiveInterpret 2.0 is an enterprise-level simultaneous interpretation model that performs real-time speech-to-speech translation with voice cloning capabilities. The model employs a duplex framework that processes input audio and generates target-language speech directly without intermediate text representations. In our evaluation, we used the official inference API with default parameters.

Despite the capability to perform direct speech-to-speech translation, current end-to-end models still face notable limitations in audio quality. In our experiments, the generated speech often contained audible artifacts or background noise, especially for longer or noisier input segments. While SeamlessExpressive improves upon previous models by attempting to clone speaker identity, prosody, and expressiveness, its expressive fidelity remains limited. The cloned voice frequently lacks the richness and nuance of natural human speech, and variations in emotion or emphasis are not always faithfully reproduced.

## C  EVALUATION METRICS

### C.1  OBJECTIVE EVALUATION

To assess the performance of speech-to-speech translation (S2ST), we report a range of objective metrics covering translation accuracy, speaker identity preservation, prosodic alignment, temporal consistency, and speech quality. For all metrics, higher values indicate better performance.

- **Text-BLEU** evaluates the fidelity of generated translations by computing corpus-level BLEU scores using the SacreBLEU library (Post, 2018). Before scoring, we apply language-specific preprocessing: English text is lowercased and stripped of punctuation (excluding apostrophes), while Chinese text is normalized to simplified characters, punctuation is removed, and characters are separated by spaces. This ensures consistency with standard BLEU evaluation practices. We use the *corpus_score* function to calculate the BLEU score across the whole dataset. Chinese samples are scored in 'zh' mode.

- **Speech-BLEU**, or ASR-BLEU, evaluates the translation fidelity of speech-to-speech translation systems. We transcribe the generated speech using ASR models, Whisper-large-v3 for English and Paraformer-zh for Chinese, and compute the BLEU score between the transcribed output and the ground-truth reference. Preprocessing follows the same pipeline as Text-BLEU.

- **AutoPCP** is an automatic implementation of the Prosodic Consistency Protocol, designed to evaluate how well the prosodic characteristics of the source speech—such as pitch contours, energy dynamics, and temporal structure—are preserved in the translated output. It extracts these features from both the input (source) and output (translated) audio and computes their similarity. By comparing the prosody between these two signals, AutoPCP provides a quantitative measure of rhythm and speaking style consistency.

- **SLC-0.2** and **SLC-0.4** (Speech Length Compliance) assess the temporal consistency between the source and translated audio by measuring whether the output duration falls within ±20% or ±40% of the original speech length. These metrics are important in real-world scenarios such as simultaneous interpretation and audiovisual dubbing, where maintaining similar speech pacing and timing is essential for naturalness, synchronization, and listener comprehension.

- **UTMOS** is a neural network-based estimator that predicts the Mean Opinion Score (MOS) for synthesized speech, serving as a proxy for human judgment of audio quality. It produces a scalar value typically ranging from 1 to 5, reflecting aspects such as naturalness, fluency, and intelligibility. This objective metric helps mitigate the subjectivity and variability of human MOS ratings, enabling reproducible and scalable quality assessment.

## C.2 SUBJECTIVE EVALUATION

To comprehensively evaluate expressive qualities, we conducted a Mean Opinion Score (MOS) listening study. Six bilingual speakers rated synthesized speech on a 5-point scale across three dimensions: emotion similarity (Emo Sim.), speaker similarity (Spk Sim.), and speech naturalness (Naturalness). Similarity is compared against the original audio instead of the synthesized audio.

We implemented the MOS evaluation using webMUSHRA (Schoeffler et al., 2018), a MUSHRA-compliant web-based audio evaluation framework that facilitates controlled listening experiments. The platform provided key features for audio assessment, including seamless audio switching and Likert scale questionnaires. Experimental configurations were defined through YAML files, with results automatically exported as CSV files. The user interface is shown in Figure C2.

In emotion similarity and speaker similarity tasks, the scale options are: "1. Definitely Different", "2. Probably Different", "3. Not be sure", "4. Probably the same", "5. Definitely the same". In speech naturalness task, the scale options are: "1. Definitely unnatural", "2. Probably unnatural", "3. Not be sure", "4. Probably natural", "5. Definitely natural".

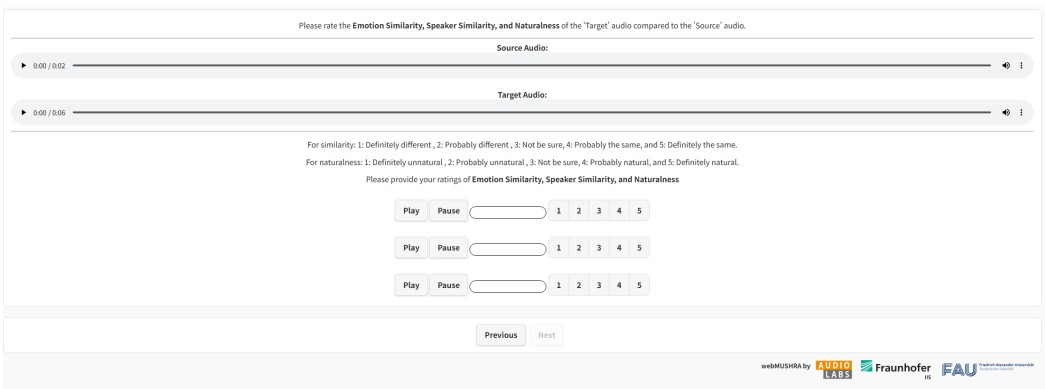

Figure C2: User interface of the webMUSHRA-based MOS evaluation system showing the three assessment dimensions: emotion similarity, speaker similarity, and speech naturalness.

# D    ADDITIONAL EXPERIMENTAL RESULTS

We also evaluate on the Chinese and English test subsets of FLEURS (Conneau et al., 2022), a massively multilingual speech dataset spanning 102 languages with n-way parallel speech and text data. UniSS demonstrates robust performance on this challenging multilingual benchmark across multiple evaluation tasks. Additionally, UniSS achieves state-of-the-art performance on speech-to-text translation (S2TT) and competitive results on automatic speech recognition (ASR) and text-to-speech synthesis (TTS) tasks.

## D.1    S2ST PERFORMANCE ON FLEURS

To verify the robustness of our framework across datasets, we further evaluate performance on the FLEURS test set. FLEURS is a multilingual benchmark derived from the FLoRes corpus that provides high-quality parallel speech and text pairs across diverse languages, making it a valuable complement to CVSS-T for assessing speech translation systems under standardized multilingual conditions.

Table D3: Main comparison results on the Fleurs dataset. Results are presented as EN-ZH | ZH-EN. Higher scores indicate better performance. '-' denotes unavailable results. Best scores are in **bold** and second-best scores are underlined.

| Category | Model | #Size | Speech-BLEU | Text-BLEU | A.PCP | SLC 0.2 | SLC 0.4 | UTMOS |
|---|---|---|---|---|---|---|---|---|
| Cascaded | 3-Stage-MT | 2.6B | 22.00 \| 21.49 | 24.56 \| 22.09 | 2.86 \| **2.73** | 0.27 \| 0.37 | 0.48 \| 0.60 | 2.14 \| 3.82 |
| | 3-Stage-LLM | 3.5B | 30.57 \| 18.45 | 33.78 \| 19.14 | **2.98** \| **2.73** | 0.50 \| 0.41 | 0.78 \| 0.64 | 2.13 \| 3.82 |
| | 2-Stage | 2.8B | 31.42 \| 23.01 | 34.72 \| 23.67 | 2.94 \| 2.06 | 0.46 \| 0.38 | 0.74 \| 0.61 | 2.13 \| 3.83 |
| MLLM | GPT-4o | - | **38.58** \| 23.21 | - \| - | 2.72 \| 2.39 | 0.45 \| 0.42 | 0.71 \| 0.64 | 3.36 \| 4.25 |
| | Qwen2.5-O | 7B | 3.37 \| 24.20 | 37.76 \| **25.15** | 1.93 \| 2.21 | 0.07 \| 0.33 | 0.27 \| 0.56 | 3.05 \| **4.33** |
| S2ST | Seamless-M | 1.2B | 14.10 \| 13.30 | 27.92 \| 17.71 | 2.19 \| 2.10 | 0.24 \| 0.15 | 0.47 \| 0.29 | 2.64 \| 3.52 |
| | Seamless-L | 2.3B | 28.86 \| 23.84 | 32.99 \| 24.06 | 2.18 \| 2.06 | 0.42 \| 0.14 | 0.66 \| 0.32 | 2.42 \| 3.90 |
| | Seamless-Ex | 1.7B | 18.80 \| 18.88 | 34.53 \| 18.89 | 2.64 \| 2.64 | 0.69 \| 0.32 | 0.88 \| 0.58 | 2.06 \| 3.19 |
| | UniSS (P) | 1.5B | 34.74 \| 22.42 | 36.12 \| 22.71 | 2.72 \| 2.65 | 0.95 \| 0.75 | 0.99 \| 0.94 | 3.40 \| 4.15 |
| | UniSS (Q) | 1.5B | 36.16 \| 23.06 | 36.96 \| 23.37 | 2.72 \| 2.64 | **0.95** \| **0.76** | **0.99** \| 0.93 | **3.41** \| 4.16 |

**Translation Fidelity.**    As shown in Table D3, UniSS (Q) achieves strong translation performance with Speech-BLEU of 36.16 and Text-BLEU of 36.96 on EN-ZH, outperforming all S2ST baselines and cascaded systems. UniSS (P) also demonstrates competitive performance across both metrics. In the ZH-EN direction, UniSS surpasses all systems with comparable model size, demonstrating the parameter efficiency of our design.

**Prosody Preservation.**    UniSS exhibits excellent prosody transfer capabilities on the FLEURS benchmark. UniSS (Q) achieves A.PCP scores of 2.72 (EN-ZH) and 2.64 (ZH-EN), outperforming all end-to-end baselines and matching GPT-4o's performance (2.72 and 2.39). Notably, UniSS achieves these results without incorporating dedicated prosody modeling modules, highlighting the effectiveness of our unified framework.

**Duration Consistency.**    Our system demonstrates exceptional speech duration consistency on the FLEURS test set. Both UniSS variants achieve near-perfect scores on the SLC-0.2 and SLC-0.4 metrics, indicating that the generated speech closely mirrors the temporal length of the input audio. This precise duration alignment is critical for real-time applications and audiovisual translation scenarios where timing mismatches can disrupt user experience.

**Speech Quality.**    UniSS achieves state-of-the-art speech quality among S2ST models. In the EN-ZH direction, UniSS (Q) attains a UTMOS score of 3.41, outperforming all open-source baselines and surpassing GPT-4o (3.36). UniSS (Q) also leads all S2ST models in the ZH-EN direction with a score of 4.16. Despite its significantly smaller size, UniSS consistently produces high-quality speech across both translation directions, confirming its effectiveness as a practical open-source solution for speech-to-speech translation.

Table D4: Performance comparison across different speech tasks. For S2TT, results are reported as EN-ZH | ZH-EN BLEU scores. Higher Text-BLEU and lower WER indicate better performance.

| Task | Dataset | Model | Metric | Result |
|------|---------|-------|--------|--------|
| S2TT | CVSS-T | Qwen2-audio | Text-BLEU ↑ | 21.86 \| 18.69 |
| | | Whisper | Text-BLEU ↑ | - \| 12.34 |
| | | UniSS | Text-BLEU ↑ | **33.17** \| **27.33** |
| | FLEURS | Qwen2-audio | Text-BLEU ↑ | 6.81 \| 17.79 |
| | | Whisper | Text-BLEU ↑ | - \| 15.65 |
| | | UniSS | Text-BLEU ↑ | **34.91** \| **22.18** |
| ASR | LibriSpeech test-clean | Whisper | WER ↓ | **1.8** |
| | | UniSS | WER ↓ | 2.4 |
| | LibriSpeech test-others | Whisper | WER ↓ | **3.6** |
| | | UniSS | WER ↓ | 6.6 |
| TTS | LibriSpeech test-clean | SparkTTS | WER ↓ \| SIM ↑ | **1.98** \| **0.584** |
| | | UniSS | WER ↓ \| SIM ↑ | 4.75 \| 0.568 |

## D.2 COMPARISON WITH S2TT SYSTEMS

Beyond speech-to-speech translation, UniSS also supports speech-to-text translation (S2TT). As shown in Table D4, UniSS achieves leading performance on this task as well. Whisper only supports X-EN translation and demonstrates poor performance on multilingual scenarios. UniSS's Text-BLEU scores significantly outperform the 7B multimodal LLM Qwen2-audio across both datasets, demonstrating superior cross-lingual speech understanding capabilities.

## D.3 ASR PERFORMANCE

Although UniSS was not specifically designed for automatic speech recognition, it demonstrates competitive ASR capabilities as shown in Table D4. We evaluate the ASR performance on the LibriSpeech dataset (Panayotov et al., 2015). On the LibriSpeech test-clean subset, UniSS achieves a WER of 2.4, which is comparable to the open-source state-of-the-art ASR model, Whisper (1.8 WER). However, on the more challenging LibriSpeech test-others subset, UniSS shows degraded performance with a WER of 6.6. It is important to note that UniSS is primarily designed as a speech-to-speech translation model, and these ASR results demonstrate its versatility beyond its core translation capabilities.

## D.4 TTS PERFORMANCE

Although UniSS was not specifically designed for text-to-speech synthesis, we evaluate its TTS capabilities on the LibriSpeech test-clean dataset as shown in Table D4. UniSS achieves a WER of 4.75 and speaker similarity of 0.568, compared to SparkTTS, which obtains 1.98 WER and 0.584 similarity. The performance gap is expected given that SparkTTS is specifically optimized for TTS, whereas UniSS is primarily designed for speech-to-speech translation. These results demonstrate that UniSS can generate intelligible speech output.

## E UNIST AND CONSTRUCTION PIPELINE

## E.1 SOURCE DATASET

We select a diverse collection of publicly available Chinese and English speech datasets as the foundation for UniSS training data. Our data sources encompass multiple domains and speaking styles to ensure robust cross-lingual speech translation capabilities.

Below is the list of source corpora we used:

- **AISHELL-3** (Shi et al., 2021): A multi-speaker Mandarin speech corpus for text-to-speech synthesis.
- **CoVoST2** (Wang et al., 2020): A multilingual speech-to-text translation corpus derived from Common Voice.
- **Common Voice (EN & ZH)** (Ardila et al., 2020): A large-scale multilingual speech corpus with speaker metadata. We use Common Voice 4.0.
- **CVSS-T** (Jia et al., 2022b): A benchmark dataset for speech-to-speech translation in English-Chinese language pairs.
- **Dailytalk** (Lee et al., 2023): A multi-speaker English speech corpus featuring conversational style recordings.
- **Emilia** (He et al., 2024): A multi-speaker multilingual speech corpus covering six languages.
- **FLEURS** (Conneau et al., 2022): A multilingual benchmark derived from FLoRes, containing parallel speech and text data.
- **Gigaspeech** (Chen et al., 2021): A large-scale multi-speaker English reading corpus.
- **Hi-Fi TTS** (Bakhturina et al., 2021): A multi-speaker English corpus with high-fidelity recordings.
- **HQ-Conversations** (Magic Data Technology Co., 2024): A multi-speaker Mandarin conversational dataset.
- **LibriSpeech** (Panayotov et al., 2015): A multi-speaker English speech corpus with reading audiobooks for TTS.
- **LibriTTS-R** (Koizumi et al., 2023): A high-quality version of the LibriTTS corpus for text-to-speech synthesis.
- **MAGICDATA** (Magic Data Technology Co., 2019): A conversational Mandarin Chinese spoken corpus.
- **NCSSD-C and NCSSD-R** (Liu et al., 2024): Multi-speaker bilingual datasets for speech synthesis.
- **VCTK** (Veaux et al., 2017): A multi-speaker English corpus featuring diverse regional accents.
- **WenetSpeech4TTS** (Ma et al., 2024): A large-scale Mandarin corpus for text-to-speech synthesis.

### E.2 TEXT TRANSLATION STEP

After cleaning the source dataset, we employ Qwen-2.5-72B-Instruct as our translation engine to translate source transcriptions into the target language. For source language Chinese audio, we use the following prompt:

```
Please translate the following text into English (Note that, aside
↪  from the translation, no other responses or explanations should
↪  be provided.):
```

For source language English audio, we use the corresponding Chinese version of this prompt to ensure consistent translation quality across language directions.

Following the translation process, we clean the translated text based on the reply format characteristics of Qwen-2.5-72B-Instruct through several steps: (1) removing common prefixes such as "Sure, here is the translation:" and its corresponding Chinese version that are automatically generated by the model; (2) eliminating explanatory notes and comments (e.g., "Note:" and corresponding Chinese version) that often accompany the translations; (3) normalizing text formatting by removing line breaks and excessive whitespace characters; and (4) filtering out mixed-language content to ensure linguistic purity, where translations containing both Chinese and English characters are discarded as invalid outputs.

### E.3 ANALYSIS OF UNIST

We conduct a subjective assessment on a randomly sampled subset of UniST to evaluate the quality of our dataset. Four raters listened to 150 examples and provided MOS scores across three dimensions: Emotion Similarity, Speaker Similarity, and Speech Naturalness. As shown in Table E5, UniST synthesized by SparkTTS achieves high speech naturalness and effectively preserves both the source speaker's voice characteristics and emotional expressiveness.

Table E5: Subjective MOS evaluation on the UniST General dataset.

| Dataset | Emo Sim.↑ | Spk Sim.↑ | Naturalness↑ |
|---|---|---|---|
| UniST General | 4.69 | 4.31 | 4.61 |

## F USE OF LLMS

Large language models are only used to polish the writing of this article without making any modifications to the content, data, or other elements. After applying LLM polishing, the authors consider the LLM outputs rather than directly copying them.

## G FREQUENTLY ASKED QUESTIONS

### G.1 WHY USE SPARK-TTS TO SYNTHESIZE THE DATASET?

Although Spark-TTS does not achieve state-of-the-art performance in objective evaluations of speaker similarity, we observe that its performance in subjective evaluations is comparable. Furthermore, we assume that its voice decoupling capability reduces the modeling complexity for large language models, which is why we deploy BiCodec for the generation speech tokenizer in the LLM. Additionally, Spark-TTS generates BiCodec tokens directly, making training more efficient by eliminating an extra encoding step.

### G.2 DOES THE PROGRESSIVE TRAINING STRATEGY INCREASE MODEL COMPLEXITY?

The progressive training strategy pertains only to the training process and does not increase the model's structural complexity. The alignment in phase 1 uses existing, easily accessible open-source data. In the ablation study section, we demonstrate the impact of different stages on model performance, proving that progressive training is effective.

### G.3 WHY DEPLOY THREE DIFFERENT SPEECH TOKENIZERS?

The speaker tokenizer is used to represent global speaker information. We hypothesize that this voice decoupling reduces the difficulty of speech translation by allowing the model to focus more on the linguistic content. We employ different linguistic and semantic tokenizers for understanding and generation to optimize each task. To generate more natural and expressive audio, the generation process uses semantic tokens, which retain information beyond the core content. However, this additional information makes content understanding more challenging. Therefore, the linguistic tokenizer, which focuses solely on content information, is better suited for the speech understanding phase. As a simple speech representation method, we believe this design does not introduce significant complexity in either training or inference.

## H REVISION LOG

Specifically, we have included the revision log here with the date format of dd/mm/yyyy (Anywhere on Earth). We welcome any suggestions that may contribute to the improvement of this paper.

### H.1 UNISS-1.1: 22/11/2025

1. Section 5.2.3: Add a new baseline 3-Stage-LLM in cascaded systems. The original 3-stage method is renamed as 3-Stage-MT.

2. Table 1 and Table D3: Add the results of new baseline 3-Stage-LLM, which performs better than 3-Stage-MT. However, it does not affect the comparison results between UniSS and cascaded systems.

3. Table 2: Add MOS results for Qwen2.5-O to demonstrate the current performance of MLLMs on the S2ST task.

