# OpenReview forum: "UniSS: Unified Expressive Speech-to-Speech Translation with Your Voice"
_ICLR.cc/2026/Conference — ICLR 2026 Poster_

### Official Review · Reviewer_9Wh8 · 2025-10-28

**Soundness:** 3
**Presentation:** 3
**Contribution:** 2
**Rating:** 4
**Confidence:** 3

**Summary:**

This work introduces UniSS, a novel single-stage framework for expressive speech-to-speech translation (S2ST) that addresses key challenges such as the scarcity of expressive paired speech data, the complexity of multi-stage pipelines, and the limited transfer of translation capabilities from large language models. UniSS integrates speech semantics and style modeling through cross-modal chain-of-thought prompting, transitioning from text-only models to text-to-speech generation. Opensourced 44.8k hours of high-quality expressive S2ST data, UniSS achieves substantial improvements in translation fidelity and speech quality, while effectively preserving voice, emotion, and duration consistency compared to prior approaches.

**Strengths:**

- Introduces a single-stage autoregressive model that directly builds on a pre-trained textual LLM, simplifying prior multi-stage architectures for expressive speech-to-speech translation.
- Provides an open-source, reproducible pipeline for constructing expressive S2ST datasets, including UniST, a large-scale Chinese-English corpus preserving speaker identity and emotional style.
- Delivers a thorough evaluation combining objective metrics (Speech/Text-BLEU, A.PCP, SLC, UTMOS) with subjective MOS assessments of emotion, speaker similarity, and naturalness.

**Weaknesses:**

- Limited language coverage: Supports only Chinese–English translation.
- Ununified tokenizer: Using separate tokenizers for linguistic and semantic tokens raises concerns about vocabulary size and token consistency.
- Language model considerations: Employing a large LLM (Qwen2.5-1.5B-Instruct) for translation may be excessive compared to traditional MT models; more efficient parameter training could better leverage other model capacities.

**Questions:**

- Is the prompt used during multi-task training also required at inference for different tasks? If so, how would it impact the available context length?

---

> ### Author Response · Authors · 2025-11-21
> **To #9Wh8**
>
> We sincerely thank you for your constructive suggestions and comments. We have conducted additional experiments to address your questions about baselines and architecture.
>
> ## 1. Weakness 1: Language Coverage
>
> We have mentioned this in the limitation section. We validated the UniSS paradigm on Chinese-English to prove the concept of using discrete tokens for expressive S2ST without architectural changes. We believe this paradigm is scalable to other languages as data becomes available.
>
> UniSS attempts to solve the S2ST task using a fully discrete token approach. Fundamentally, we treat speech as a "new language" in text format, enabling the LLM to understand and generate it by easily expanding the codebook. UniSS validates that it is possible to build a Speech LLM using purely discrete representations without modifying the backbone LLM architecture or relying on complex two-stage structures.
>
> ## 2. Weakness 2: Triplet Speech Tokenizer Design
>
> Different speech tokenizers are deployed to model different dimensions of speech (semantic and acoustic details). Creating a single tokenizer for both high-quality understanding and generation is a recognized open challenge [1,2], often resulting in massive parameter counts and complex training. Our design effectively circumvents this bottleneck by using specialized discrete tokens, making the model more reproducible.
>
> ## 3. Weakness 3: Efficient Parameters and Baseline Fairness
>
> Thank you for the suggestion regarding parameter efficiency and fair comparison. We agree that using LoRA or similar efficient parameter training methods is a reasonable approach that can reduce training complexity.
>
> To ensure a rigorous comparison, we replaced the traditional MT model (NLLB-600M) in the 3-stage cascaded baseline with the larger **Qwen2.5-1.5B-Instruct** (matching our backbone). We evaluated UniSS against two new strong cascaded pipelines:
>
> a. Original ASR + **Qwen2.5-1.5B-Instruct** (New MT) + Cosyvoice 2 (Original TTS).
>
> b. Original ASR + **Qwen2.5-1.5B-Instruct** (New MT) + SparkTTS (New TTS).
>
> | Model | # Size | BLEU-Speech | BLEU-Text | SLC 0.2 | SLC 0.4 | UTMOS |
> | --- | --- | --- | --- | --- | --- | --- |
> | Stage-3 a | 3.5B | 26.74 $\vert$ 17.69 | 28.36  $\vert$ 18.51 | 0.60  $\vert$ 0.48 | 0.87  $\vert$ 0.84 | **3.76**  $\vert$ 3.48 |
> | Stage-3 b | 3.5B | 26.70  $\vert$ 18.20 | 28.39  $\vert$ 18.65 | 0.48  $\vert$ 0.47 | 0.76  $\vert$ 0.80 | 3.71  $\vert$ 3.27 |
> | UniSS (Q) | 1.5B | **32.20**  $\vert$ **24.28** | **32.95**  $\vert$ **26.28** | **0.98**  $\vert$ **0.87** | **0.99**  $\vert$ **0.97** | **3.76**  $\vert$ **3.86** |
>
> The results demonstrates that UniSS continues to outperform these enhanced cascaded systems, even when they utilize a LLM for translation and advanced TTS modules. We also add the entire table in Appendix for comparison.
>
> ## 4. Question 1: Prompt Consistency and Context Length
>
> We thank for your insightful question. Yes, the prompts used during training and inference are consistent. We use special tokens (e.g., <|task_tts|>, <|slow_mode|>) to guide the task, and some other special tokens are used to instruct speed and languages.
>
> In a single inference pass, the prompt and output structure contain approximately 22 special tokens (including task, language, duration control, and start/end tokens). Given that speech token sequences are typically much longer, this overhead has a negligible impact on the available context length.
>
> ## Reference
>
> [1] Yan, Canxiang, et al. "Ming-UniAudio: Speech LLM for Joint Understanding, Generation and Editing with Unified Representation." *arXiv preprint arXiv:2511.05516* (2025).
>
> [2] LLM-Core-Team Xiaomi. "MiMo-Audio: Audio Language Models are Few-Shot Learners."

---

### Official Review · Reviewer_41XM · 2025-10-31

**Soundness:** 3
**Presentation:** 3
**Contribution:** 3
**Rating:** 4
**Confidence:** 5

**Summary:**

The paper presents **UniSS**, a unified single-stage speech-to-speech translation (S2ST) system designed to translate spoken content while preserving both speaker identity and expressive emotional style. UniSS eliminates multi-stage pipelines by leveraging a pretrained large language model (LLM) backbone, incorporating speech and text with a shared token vocabulary, and introducing cross-modal chain-of-thought (CoT) prompting that transfers text translation capabilities to the speech domain. Besides model proposals, the authors construct UniST, a large-scale expressive S2ST dataset (44.8k hours). Experimental results on several benchmarks demonstrate substantial improvements in translation fidelity, expressiveness, and runtime efficiency versus established baselines.

**Strengths:**

1. UniSS is a conceptually clean, single-model approach that collapses ASR, MT, and TTS into one autoregressive LLM with expanded vocabulary for both text and discrete speech tokens.
2. The construction of UniST (44.8k hours) as a high-quality, diverse Chinese-English corpus with explicit control over speaker, emotion, and timing, is a notable resource for the field.

**Weaknesses:**

1. The subjective evaluation does not include MLLMs such as GPT-4o, and the model’s performance is also inferior to Seed Live or Seamless-Ex on certain metrics.

2. I don’t think the so-called cross-modal prompting can be regarded as a form of Chain of Thought (CoT).


3. Despite its claim of being “unified,” UniSS relies on three distinct speech tokenizers (speaker, linguistic, and semantic), each originating from separate models. This design increases the vocabulary size (over 180k tokens) and may limit scalability, efficiency, or ease of deployment when supporting additional languages or larger vocabularies.

**Questions:**

1. In Phase 1, is the TTS task really useful? After all, your ultimate goal is translating source speech to target speech.

2. How reliable is the metric A.PCP?

---

> ### Author Response · Authors · 2025-11-21
> **To #41XM**
>
> We sincerely thank you for your constructive suggestions and comments. We have addressed your concerns regarding the comparisons, definitions, and model architecture below.
>
> ## 1. Weakness 1: Subjective Evaluation and Comparisons
>
> We have added subjective evaluation results for **Qwen2.5-Omni**. Please note that Qwen2.5-Omni and GPT-4o Audio utilize a default voice and do not support zero-shot voice cloning, so Speaker Similarity is not applicable. Results show that the MLLM struggles with emotion consistency, while its naturalness is comparable to UniSS, despite Qwen2.5-Omni having more than 4x the parameters.
>
> | Model  | Emo Sim.↑ | Spk Sim.↑ | Naturalness↑ |
> | --- | --- | --- | --- |
> | UniSS (Q) | 4.51  | 4.42  | 4.45 |
> | Qwen2.5-O | 2.84 | N/A | 4.47 |
>
> Regarding **Seamless-Ex**: UniSS outperforms Seamless-Ex on most metrics. The only exception is A.PCP (prosody), for which UniSS was not explicitly optimized, whereas Seamless-Ex uses specific data and a PRETSSEL vocoder to guarantee it. Note, UniSS still surpasses MLLMs and Seamless-Large on A.PCP.
>
> Regarding **Seed Live**: This is a closed-source commercial system. While it sets a high bar, UniSS achieves comparable or superior performance on evaluated metrics, which is significant for an open academic contribution.
>
> ## 2. Weakness 2: Definition of Cross-Modal CoT
>
> We frame the Listen-Translate-Speak sequence as a reasoning chain. Unlike direct S2ST, this intermediate step decomposes the complex task, significantly improving translation accuracy. While distinct from purely textual CoT, it serves the analogous function of "thinking" (understanding and translating text) before "acting" (speaking), hence the terminology.
>
> ## 3. Weakness 3: Unified Model vs. Multiple Tokenizers
>
> Using distinct tokenizers allows for specialized modeling of different speech dimensions (speaker, linguistics, semantics) without compromising the "unified" nature of the model—we still process all modalities via a single LLM backbone (similar to approaches like Bagel [1] or Janus-Pro [2] which use "unified" in their titles).
>
> Creating a single tokenizer for both high-quality understanding and generation is a recognized open challenge [3,4], often resulting in massive parameter counts and complex training. Our design effectively circumvents this bottleneck by using specialized discrete tokens, making the model more reproducible.
>
> UniSS attempts to solve the S2ST task using a fully discrete token approach. Fundamentally, we treat speech as a "new language" in text format, enabling the LLM to understand and generate it by easily expanding the codebook. UniSS validates that it is possible to build a Speech LLM using purely discrete representations without modifying the backbone LLM architecture or relying on complex two-stage structures.
>
> ## 4. Question 1: Utility of TTS in Phase 1
>
> Thanks for your insightful question. Conceptually, our strategy employs **curriculum learning**:
>
> - **Stage 1:** Completes simpler tasks. ASR and TTS tasks align speech-text modalities (handling understanding and generation, respectively). The S2TT task warms up the model for translation, while the MT task prevents catastrophic forgetting of text capabilities.
> - **Stage 2 & 3:** The model learns S2ST, instruction following for different modes, and duration consistency.
>
> Empirical evidence highlights the critical role of Stage 1: When training Stage 2 directly with 2,100 hours of data, the Quality Mode achieves a Speech-BLEU of 24.97 (surpassing many baselines). This result confirms the necessity of the pre-alignment in Stage 1. With the full UniST data, this value improves to 28.24. Without TTS task trained in stage 1, it is difficult for UniSS to master generation capability in S2ST task.
>
> ## 5. Question 2: Reliability of A.PCP
>
> A.PCP is an established metric introduced by Meta in Seamless [5] and utilized in recent top-tier works like TransVIP [6]. It is not a metric proposed by us. As noted in Response 1. Weakness1, UniSS performs competitively even without specific optimization for this metric.
>
> ## Reference
>
> [1] Deng, Chaorui, et al. "Emerging properties in unified multimodal pretraining." *arXiv preprint arXiv:2505.14683* (2025).
>
> [2] Chen, Xiaokang, et al. "Janus-pro: Unified multimodal understanding and generation with data and model scaling." *arXiv preprint arXiv:2501.17811* (2025).
>
> [3] Yan, Canxiang, et al. "Ming-UniAudio: Speech LLM for Joint Understanding, Generation and Editing with Unified Representation." *arXiv preprint arXiv:2511.05516* (2025).
>
> [4] LLM-Core-Team Xiaomi. "MiMo-Audio: Audio Language Models are Few-Shot Learners."
>
> [5] Barrault, Loïc, et al. "SeamlessM4T: massively multilingual & multimodal machine translation." *arXiv preprint arXiv:2308.11596* (2023).
>
> [6] Le, Chenyang, et al. "Transvip: Speech to speech translation system with voice and isochrony preservation." *Advances in Neural Information Processing Systems* 37 (2024): 89682-89705.

---

### Official Review · Reviewer_yrvR · 2025-11-01

**Soundness:** 3
**Presentation:** 3
**Contribution:** 3
**Rating:** 6
**Confidence:** 4

**Summary:**

The paper introduces UniSS, a single-stage autoregressive framework for expressive speech-to-speech translation (S2ST) that preserves speaker voice, emotion, and duration while achieving high translation fidelity. Built on the pre-trained Qwen2.5-1.5B-Instruct LLM, it uses a triple-tokenizer strategy (speaker, linguistic, semantic tokens) and cross-modal CoT prompting to transfer text translation capabilities to speech. Two modes (Quality and Performance) balance fidelity and efficiency. The authors release UniST, a 44.8k-hour Chinese-English expressive S2ST dataset. Experiments on CVSS-T and FLEURS show superior performance over baselines like SeamlessM4T, GPT-4o, and cascaded systems in metrics such as Speech-BLEU, prosody consistency, duration compliance, and speech quality. Subjective MOS evaluations confirm high emotion/speaker similarity and naturalness.

**Strengths:**

1. The paper addresses an important problem in speech-to-speech translation: preserving the speaker's voice, emotional style, and prosody across languages. This is crucial for applications like multilingual communication or media dubbing. The motivation is clear, and the cross-modal CoT approach effectively aligns text-based LLM capabilities with speech in a single model. This method is innovative and necessary, as ablations show that removing CoT leads to major drops in performance.

2. The authors release UniST, a 44.8k-hour dataset for expressive S2ST, which tackles the key issue of limited paired data with style preservation. The creation pipeline is well-documented, making it valuable for the research community.

3. The experiments cover a broad set of metrics, including semantic and acoustic part. Baselines are varied, spanning cascaded systems and end-to-end models, ensuring robust comparisons.

4. UniSS outperforms baselines on most metrics, such as emotion similarity and duration matching. The demopage also illustrates the model's success in maintaining expressive qualities.

5. The paper is well-structured and readable, with clear descriptions of the architecture, training strategy, and ablations that validate the approach.

**Weaknesses:**

1. The novelty is limited. The model's strength in maintaining speaker voice, emotion, and prosody largely relies on pre-existing components, such as the BiCodec tokenizer for speaker representations and SparkTTS for synthesizing expressive data. UniSS itself mainly integrates these by organizing a multiple token sequence then inputing it into a LLM, reducing its core innovation and novelty.

2. The "Single-Stage" claim is misleading.  While positioned as a streamlined single-stage model to avoid cascaded errors and latency, the Quality mode generates intermediate source text (T_src) and target text (T_tgt) before speech tokens, mirroring the sequential steps of a traditional 3-stage pipeline (ASR → MT → TTS). This undermines the claimed advantages and could lead to similar error accumulations and delays.

3. Training and evaluations are centered on English-Chinese pairs. Besides, the use of GLM tokenizer and SparkTTS also constrains the method for applying on other languages. This raises questions about scalability to diverse languages, accents, or noisy real-world scenarios, where data scarcity for expressiveness could amplify issues.

4. A significant issue is the unfair setting in evaluations, particularly for translation accuracy (BLEU). The UniST training data uses high-quality translations from the large Qwen2.5-72B model, while the 3-stage cascaded baseline relies on the much smaller pretrained NLLB-600M for machine translation. Why not use Qwen-2.5-1.5B-Instruct LLM in the cascaded system?

5. The framework's heavy reliance on text-based elements. For example, CoT prompting with intermediate transcripts makes it unsuitable for unwritten or low-resource languages without scripts. This contradicts goals in textless S2ST research and should be highlighted as a key limitation, especially given the focus on expressive translation.

**Questions:**

1. To make a fair comparison, it will be better if you test another cascaded baseline that uses the same Qwen-2.5-1.5B-Instruct model for MT and the same SparkTTS model for TTS. Besides, consider training the baselines on the UniST dataset. This would show if the UniSS framework is truly better, or if it just benefited from better components and training data.

2. How does the cross-modal CoT prompting specifically bridge text and speech modalities? Provide some studies of intermediate outputs (e.g., T_src, T_tgt) to explain whether UniSS addresses error propagation problems.

3. The motivation emphasizes unifying expressiveness, but much relies on existing tools (BiCodec, SparkTTS). What novel insights does UniSS offer for LLM-based S2ST, and how does it differ from prior CoT adaptations in multimodal tasks?

4. For training strategy, what is the individual contribution of each task to overall performance? Could you add ablations removing or weighting specific tasks to quantify their effects?

5. Since UniST is fully a synthetic dataset generated by SparkTTS, is it unable to effectively capture word-level semantic prosody mapping or relevance in speech translation? For instance, the transfer of semantic nuances such as emphasized stress. It is recommended to provide corresponding analysis and discuss the potential limitations.

---

> ### Author Response · Authors · 2025-11-21
> **To #yrvR (Part 1)**
>
> We sincerely thank you for your constructive suggestions and positive comments. Your insights regarding baseline selection and error propagation were particularly helpful. Following your suggestions, new experiments and ongoing extensions are illustrated here and in the Appendix for the benefit of the community.
>
>
> ## 1. Question1 & Weakness4: Baselines and Fairness of Comparison
>
> To address the concern regarding fair comparison, we evaluated UniSS against two new strong cascaded pipelines:
>
>   a. Original ASR + **Qwen2.5-1.5B-Instruct** (New MT) + Cosyvoice 2 (Original TTS).
>
>   b. Original ASR + **Qwen2.5-1.5B-Instruct** (New MT) + SparkTTS (New TTS).
>
> The results, presented in table below (and fully detailed in the Appendix), demonstrate that UniSS continues to outperform these enhanced cascaded systems, even when they utilize a LLM for translation and advanced TTS modules.
>
> | Model | # Size | BLEU-Speech | BLEU-Text | SLC 0.2 | SLC 0.4 | UTMOS |
> | --- | --- | --- | --- | --- | --- | --- |
> | Stage-3 a | 3.5B | 26.74 $\vert$ 17.69 | 28.36  $\vert$ 18.51 | 0.60  $\vert$ 0.48 | 0.87  $\vert$ 0.84 | **3.76**  $\vert$ 3.48 |
> | Stage-3 b | 3.5B | 26.70  $\vert$ 18.20 | 28.39  $\vert$ 18.65 | 0.48  $\vert$ 0.47 | 0.76  $\vert$ 0.80 | 3.71  $\vert$ 3.27 |
> | UniSS (Q) | 1.5B | **32.20**  $\vert$ **24.28** | **32.95**  $\vert$ **26.28** | **0.98**  $\vert$ **0.87** | **0.99**  $\vert$ **0.97** | **3.76**  $\vert$ **3.86** |
>
> Regarding the training data: We believe the UniST dataset unlocks significant training potential. However, the training data used by other state-of-the-art baselines is not inferior. Public MLLMs (e.g., Qwen2.5-Omni) and closed-source models (e.g., GPT-4o Audio, Seed Live Interpret 2.0) are trained on high-quality, massive datasets. For cascaded systems, which benefit from extensive, high-quality training data across their specific sub-modules (ASR/MT/TTS), fine-tuning on UniST offers marginal value as they are already highly optimized for their specific tasks.
>
> ## 2. Question2 & Weakness2: Intermediate Text and Error Propagation
>
> Your insight regarding intermediate text generation is correct; it does influence the translation accuracy. To quantify this, we conducted three oracle experiments: replacing the source text with Ground Truth (GT-T_src), replacing the target text (GT-T_tgt), and replacing both.
>
> Table below shows that the performance gap between **UniSS (Quality Mode)** and the **GT-T_src** setting is minimal. This indicates that the "Listen" step does not introduce significant error. The primary performance gains observed in the **GT-T_tgt** setting suggest that the bottleneck lies within the translation capability itself. Furthermore, comparing UniSS to cascaded systems confirms that our paradigm exhibits significantly lower error propagation than traditional three-stage pipelines.
>
> | Setting | Speech-BLEU | Text-BLEU |
> | --- | --- | --- |
> | UniSS (Q)  | 32.20  $\vert$ 24.28 | 32.95  $\vert$ 26.28 |
> | GT-T_src | 32.74  $\vert$ 24.59 | 35.26  $\vert$ 27.02 |
> | GT-T_tgt | 86.41  $\vert$ 82.82 | N/A |
> | GT-T_src + GT-T_tgt | 86.41  $\vert$ 82.90 | N/A |
>
> ## 3. Question 3 & Weakness 1: Novelty and Innovation
>
> UniSS attempts to solve the S2ST task using a fully discrete token approach. Fundamentally, we treat speech as a "new language" in text format, enabling the LLM to understand and generate it by easily expanding the codebook.
>
> - **Feasibility:** UniSS validates that it is possible to build a Speech LLM using purely discrete representations without modifying the backbone LLM architecture or relying on complex two-stage structures.
> - **Tokenizer Unification:** While we use multiple tokenizers, a single tokenizer capable of unifying both *understanding* and *generation* remains an open research challenge [1,2]. Our strategy effectively bridges this gap using existing tools.
> - **Cross-Modal CoT:** Unlike prior S2ST work, which often lacks intermediate text (T_src) or flexible modes, our Cross-Modal Chain-of-Thought uses text tokens as a pivot to leverage the pretrained LLM's textual capabilities. To our knowledge, UniSS is the first work to achieve expressive S2ST using fully discrete tokens without structural modification to the LLM.

---

> ### Author Response · Authors · 2025-11-21
> **To #yrvR (Part 2)**
>
> ## 4. Question 4: Training Strategy Ablation
>
> Thanks for your valuable suggestion. However, due to the extensive computational resources required for Stage 1 pre-training, we cannot re-run the full ablation during the short rebuttal period.
>
> Conceptually, our strategy employs **curriculum learning**:
>
> - **Stage 1:** Completes simpler tasks. ASR and TTS tasks align speech-text modalities (handling understanding and generation, respectively). The S2TT task warms up the model for translation, while the MT task prevents catastrophic forgetting of text capabilities.
> - **Stage 2 & 3:** The model learns S2ST, instruction following for different modes, and duration consistency.
>
> Empirical evidence highlights the critical role of Stage 1: When training Stage 2 directly with 2,100 hours of data, the Quality Mode achieves a Speech-BLEU of 24.97 (surpassing many baselines). This result confirms the necessity of the pre-alignment in Stage 1. With the full UniST data, this value improves to 28.24.
>
> ## 5. Question5: Word-level Semantic Prosody
>
> We thank for this insightful question. We did not explicitly design for word-level prosody alignment (e.g., specific stress or pauses). However, we observe that the model exhibits implicit alignment capabilities (as seen in the demo samples). We hypothesize two reasons:
>
> 1. The semantic and linguistic tokens inherently retain prosodic information.
> 2. The short audio segments used in training facilitate learning tight alignments, similar to word-level mapping, which generalizes to longer sequences.
>
> ## 6. Weakness 3: Multilingual Support
>
> We have mentioned this in the limitation section. We validated the paradigm on Chinese-English due to high-quality data availability. We believe this framework is scalable to other languages. As speech tokenizer technology matures, the UniSS paradigm is likely to achieve even better performance across diverse languages.
>
> ## 7. Weakness 5: Reliance on Text
>
> We acknowledge this limitation. While textless S2ST is an important research frontier, current solutions struggle with accuracy. Most of the well-performing jobs today rely on written communication [3,4]. Incorporating intermediate text significantly boosts translation fidelity, making the system more viable for real-world applications.
>
> ## Reference
>
> [1] Yan, Canxiang, et al. "Ming-UniAudio: Speech LLM for Joint Understanding, Generation and Editing with Unified Representation." *arXiv preprint arXiv:2511.05516* (2025).
>
> [2] LLM-Core-Team Xiaomi. "MiMo-Audio: Audio Language Models are Few-Shot Learners."
>
> [3] Labiausse, Tom, et al. "High-fidelity simultaneous speech-to-speech translation." *arXiv preprint arXiv:2502.03382* (2025).
>
> [4] Le, Chenyang, et al. "Transvip: Speech to speech translation system with voice and isochrony preservation." *Advances in Neural Information Processing Systems* 37 (2024): 89682-89705.

---

### Author Response · Authors · 2025-11-23
**Thanks to the insightful and professional reviewers.**

Dear all reviewers,

We sincerely appreciate your constructive comments and the time you dedicated to reviewing our work. We are encouraged by your recognition of the UniSS framework's innovation, the effectiveness of our cross-modal CoT, and the significant value of the 44.8k-hour UniST dataset to the community.

In response to your insightful questions and valuable suggestions, we have dedicated considerable effort over the past few days to conducting new experiments and incorporating your advice. Specifically, we have enhanced the following aspects in the revised draft:

- More Rigorous Baselines: To address concerns regarding fairness (#yrvR, #9Wh8), we have implemented and evaluated two new strong cascaded baselines. These utilize the same LLM backbone (Qwen2.5-1.5B-Instruct) and advanced TTS models (SparkTTS, Cosyvoice 2) as our method. The results (Table 1 in new version paper) confirm that UniSS continues to achieve superior performance even against these enhanced cascaded systems.
- In-depth Analysis & MLLM Comparison: We conducted oracle experiments to quantify the impact of intermediate text, verifying the low error propagation of our "Listen-Translate-Speak" paradigm (#yrvR). Additionally, we added subjective evaluations comparing UniSS with state-of-the-art MLLMs (Qwen2.5-Omni), demonstrating our advantages in emotion consistency (#41XM).
- Clarifications & Writing: We have thoroughly clarified the definitions of "Cross-modal CoT" and "Single-Stage" (#41XM, #yrvR), and provided a detailed justification for our multi-tokenizer design (#9Wh8). We also response the questions about multilingual scalability and text reliance.

We have modified our paper based on your  suggestions and included a detailed revision log in the Appendix H. We believe these additional experiments and revisions strongly support the validity and contribution of UniSS. We welcome any further feedback from you.

---

### Meta-Review · Area_Chair_4hN9 · 2026-01-03

**Summary:**

The paper presents an end-to-end speech translation system, dubbed UniSS, that focuses on conserving not only the semantics but style of the speech (e.g., voice, emotion, duration consistency). The authors also build and publicly release UniST, 44.8K hour syntehsizeed dataset of expressive S2ST. The proposed method is built on the pre-trained Qwen2.5-1.5B-Instruct LLM, and introduces a triple-tokenizer strategy (speaker, linguistic, semantic tokens) and cross-modal CoT prompting to transfer text translation capabilities to speech.

The reviewer pointed out following core weaknesses: (1) limited experiment scope (only covering English-Chinese language pair), (2) over claiming about “unified” framework given separate tokenizer for linguistic / semantic tokens and (3) fairness over baselines, given this was trained with high-quality translations from the Qwen2.5-72B model. While weakness (1) is a real limitation (and I’d recommend authors should mention this in the abstract clearly), the weakness (3) is mostly addressed with author’s new baseline experimental results. The AC does not think (2) is necessarily big downside of the approach itself.

The single model approach that combines ASR, MT, TTS into one autoregressive LLM is novel and impressive, and its effectiveness is well demonstrated with competitive baselines and diverse metrics (that captures both acoustic / translation qualities).

**Reviewer Concerns:**

See the meta review above.

**Reviewer Scores:**

I think reviewer yrvR would have increase score, given the authors have provided new experimental results that addresses many aspects of reviewers concern and new ablation study. I think reviewer 41XM might have increased a score a bit (new subjective evaluation comparison results) but could have kept the original score. I do not think other reviewers would have changed the score.

---

### Decision · Program_Chairs · 2026-01-26

Accept (Poster)